# Tracing development of song memory with fMRI in zebra finches after a second tutoring experience

Payal Arya[1], Stela P. Petkova[1], Praveen P. Kulkarni[2], Nancy H. Kolodny[3] & Sharon M. H. Gobes [1✉]

Sensory experiences in early development shape higher cognitive functions such as language acquisition in humans and song learning in birds. Zebra finches (*Taeniopygia guttata*) sequentially exposed to two different song 'tutors' during the sensitive period in development are able to learn from their second tutor and eventually imitate aspects of his song, but the neural substrate involved in learning a second song is unknown. We used fMRI to examine neural activity associated with learning two songs sequentially. We found that acquisition of a second song changes lateralization of the auditory midbrain. Interestingly, activity in the caudolateral Nidopallium (NCL), a region adjacent to the secondary auditory cortex, was related to the fidelity of second-song imitation. These findings demonstrate that experience with a second tutor can permanently alter neural activity in brain regions involved in auditory perception and song learning.

[1] Neuroscience Department, Wellesley College, Wellesley, MA 02481, USA. [2] Center for Translational Neuroimaging, Northeastern University, Boston, MA 02115, USA. [3] Chemistry Department, Wellesley College, Wellesley, MA 02481, USA. ✉email: sgobes@wellesley.edu

During sensitive periods in development, external stimuli shape behavior by stimulating the brain to learn from and adapt to changing environments[1]. Language is one example of a skill learned during a sensitive period, and neural plasticity enables the acquisition of a second language[2,3]. Neuroimaging studies have revealed that activation in Broca's and Wernicke's areas of the left hemisphere overlaps for both first and second language, which suggests a common cortical network for second language perception and production[4–8].

Similar to speech acquisition in humans, song learning in zebra finches occurs during a sensitive period early in development[9,10]. Juvenile male zebra finches usually learn from a single adult male, the "tutor", and form an auditory memory of his song during the sensory phase of learning[11–14]. Once they have started to imitate their father's song, zebra finches can learn the song of a second tutor if it is introduced during their sensitive period[15,16]. Like the acquisition of multiple languages in humans[17–19], second-song learning in zebra finches requires plasticity in brain regions involved in song learning to accommodate new information. Neurons in the lateral magnocellular nucleus of the anterior Nidopallium (lMAN) respond to auditory playbacks of the first tutor song in juveniles, but once the birds have learned from their second tutor those responses are lost or overwritten[16]. In a region functionally analogous to Wernicke's area in humans, the caudomedial Nidopallium (NCM, part of the caudal Nidopallium [NC] in Fig. 1a), the immediate early gene (IEG) response to tutor song is left-dominant in zebra finches reared with only one tutor[20]. When zebra finches are tutored sequentially by two different adult conspecifics, a left-dominant IEG response in the NCM correlates with better learning of the song from the second tutor, whereas a right-dominant IEG response correlates with better song learning from the first tutor[21]. This raises the possibility that neural representations of multiple auditory memories can be stored in the brain simultaneously, but that acquisition of a second song permanently alters the neural substrate for perception and memory of vocalizations.

To investigate the learning-related changes that accompany the acquisition of a novel song memory, we used blood-oxygen-level-dependent (BOLD) functional MRI. fMRI was performed in developing male zebra finches at two different timepoints, before (55 days post hatching [dph]) and after exposure to a second tutor (90 dph), to elucidate experience-dependent changes of learning a new song at the level of the entire brain. Our results indicate that experience with a second tutor during the sensorimotor phase of learning alters neural activity in the auditory midbrain. In addition, they suggest that the caudolateral Nidopallium (NCL), a region adjacent to the secondary auditory region (NCM) and analogous to the mammalian prefrontal cortex, is involved in higher-order processing of tutor songs in the avian brain.

## Results

**Zebra finches can successfully learn songs from two sequential tutors.** We raised male zebra finches with either a single tutor (Control Group) or two different tutors (Sequentially Tutored Group), with the first tutoring period from 0 to 32 dph and the second tutoring period from 55 to 65 dph (Fig. 1b). To determine if the birds had imitated their tutors' songs, we analyzed the similarity between the tutee's song and the tutor's song at 55 dph and 90 dph, by determining the percentage of shared elements (see "Methods"). Although juvenile songs at 55 dph are not yet stereotyped, there was already significant similarity between the tutees' and tutors' (TUT1) songs as compared to novel conspecific songs (Fig. 1c: Control Group: $TUT1_{55dph}$ vs $NOV_{55dph}$: $Z = 3.4$, $P = 0.02$; Sequentially Tutored Group: $TUT1_{55dph}$ vs $TUT2_{55dph}$: $Z = 2.19$, $P = 0.02$). Thus, the tutees already copied parts of the song of their first tutor at 55 dph before re-exposure to that tutor or the introduction of a second tutor.

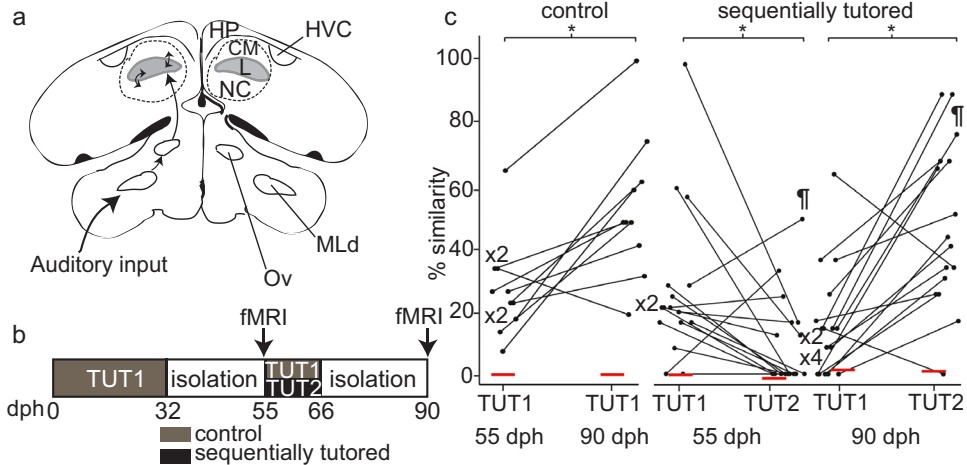

**Fig. 1 Anatomy of the avian brain, experimental timeline, and song learning in control and sequentially tutored birds. a** Schematic showing the avian auditory pathway in coronal plane. Pathway indicated in the left hemisphere, with identical regions labeled in the right hemisphere. Line drawings of a coronal section adapted from the zebra finch stereotaxic atlas[89]. MLd: dorsal lateral nucleus of the mesencephalon lateralis dorsalis, NC: caudal nidopallium (includes both the caudomedial nidopallium [NCM] and the caudolateral nidopallium [NCL]), CM: caudal mesopallium (includes the caudomedial mesopallium [CMM]), HP: hippocampus, Ov: nucleus ovoidalis, L: field L. **b** Experimental timeline. Juvenile male zebra finches were housed with their first tutor (TUT1) until 32 dph and then individually isolated in acoustic chambers. From 55–66 dph, juveniles were either housed together with their TUT1 (control) or a novel tutor (TUT2), and subsequently isolated and sacrificed at 150 dph. Two fMRI sessions were performed: at 55 dph and at 90 dph (black arrows). **c** Zebra finches can learn to imitate songs from two tutors sequentially. % Similarity is expressed as the percentage of shared elements between the tutee's song and the tutor's song. * indicates significant differences ($P < 0.05$) in similarity scores for 55 vs. 90 dph in control birds (left, $N = 10$), and for TUT1 vs TUT2 in 55 dph (center) and 90 dph (right) sequentially tutored birds ($N = 16$). ¶ indicates a significant increase ($P < 0.05$) in similarity scores with the second tutor between 55 dph and 90 dph. Each dot and line represent an individual bird. Red horizontal lines indicate mean similarity with an entirely different set of unfamiliar songs (different from those used as second tutors, which are also novel at 55 dph).

At 90 dph, the songs from the control group, exposed to the same tutor from 0 to 32 dph and again from 55 to 65 dph, resembled the tutors' songs more than the novel conspecific songs (Fig. 1c: TUT1$_{90dph}$ vs NOV$_{90dph}$: $Z = 2.8$, $P = 0.02$). In contrast, sequentially tutored birds had copied most of the syllables from their second tutor and showed less similarity with TUT1 songs (Fig. 1c: TUT1$_{90dph}$ vs TUT2$_{90dph}$: $Z = -2.9$, $P = 0.003$). The resemblance to TUT2 songs was significantly different from random similarity to unrelated birds ($Z = 3.4$, $P = 0.0004$). Over the course of development, songs from birds exposed to only one tutor (control) became better imitations of the tutors' songs (TUT$_{55dph}$ vs TUT$_{90dph}$; $Z = -2.7$, $P = 0.007$), whereas for sequentially tutored birds there was a significant increase in similarity scores with the second tutor (TUT2$_{55dph}$ vs TUT2$_{90dph}$; $Z = -2.6$, $P = 0.009$). These results are consistent with previous studies that show that sequentially tutored birds can imitate (part of) the song from a second song tutor[15,16,21].

**Novel song induces stronger activity in the right anterior forebrain (lMAN) during early sensorimotor development as compared to learned song**. To investigate whether there was learning-related neural activity in juvenile zebra finches, we compared the BOLD responses to playback of TUT1 and playback of NOV in 55-day-old birds. At that point, all animals ($N = 28$) were reared with their family (other nestlings and mother) and with their biological father as the first tutor (TUT1) until 32 dph. At 55 days old, the voxel-based analysis revealed activation of parts of the auditory lobule (the primary auditory region field L and the caudomedial Mesopallium [CMM]), and the caudolateral Nidopallium (NCL)[22] in both hemispheres in response to TUT1 (Fig. 2a: TUT1 > Rest, exploratory threshold $P_{uncorrected} < 0.05$, one-sample $t$ test). In response to novel conspecific songs, there was activation of large parts of the auditory lobule (including field L, the CMM, and the caudomedial Nidopallium [NCM]), the NCL, and the lateral magnocellular nucleus of the anterior Nidopallium (lMAN) (Fig. 2b: NOV > Rest, exploratory threshold $P_{uncorrected} < 0.05$, one-sample $t$ test). Only in the right hemisphere, there was response specificity in a cluster that enclosed lMAN core and shell for novel conspecific songs over TUT1 songs (Fig. 2d: NOV > TUT1: $t_{max} = 4.8$, $P_{uncorrected} < 0.001$, paired $t$ test). We confirmed with ROI analysis that BOLD activity in lMAN was specific for NOV songs (Fig. 2f, g: ROI analysis; $t_{max} = 4.3$, $P_{FWE} = 0.004$), and that BOLD activity in Area X approached significance ($t_{max} = 3.8$, $P_{FWE} = 0.06$). On closer inspection of the BOLD response in lMAN, a higher positive BOLD response to NOV songs was also accompanied by a negative BOLD response to TUT songs (Fig. 2g). Thus, at 55 dph, when juveniles have started to learn from one song tutor, the anterior forebrain region lMAN is activated more strongly by novel conspecific songs as compared to learned songs (TUT1).

**Hemispheric differences in auditory midbrain activity depend on the learning experience**. In sequentially tutored birds, we found a pronounced effect of playback of songs in the right midbrain and hindbrain (Fig. 3a: supra threshold clusters from one-way ANOVA; midbrain cluster: $F_{max} = 9.12$, $P_{uncorrected} = 0.001$; hindbrain cluster: $F_{max} = 9.32$, $P_{uncorrected} = 0.001$). Post hoc tests confirmed that the right midbrain and hindbrain clusters were significantly activated by TUT2 song (Fig. 3a: TUT2 > TUT1: midbrain cluster: $t_{max} = 4.08$, $P_{uncorrected} = 0.0002$; hindbrain cluster: $t_{max} = 4.08$, $P_{uncorrected} = 0.0002$). A repeated measures ANOVA between 55- and 90-day-old birds confirmed that TUT2 song-specific neural responses in the mid/hindbrain were altered after the experience with the second tutor (Supplementary Fig 1b: TUT2$_{90 vs 55 dph}$, $t_{max} = 3.42$,

$P_{FWE} = 0.024$). Interestingly, learning-related activity in the right midbrain/hindbrain was specific to adult sequentially tutored birds whereas, in the control group, higher BOLD activity was found in the left midbrain/hindbrain regions in response to learned song over novel song (Fig. 3f: TUT > NOV: $t_{max} = 4.04$, $P_{uncorrected} = 0.001$). To further investigate the activated cluster in the midbrain of control and sequentially tutored birds, which included the dorsal lateral nucleus of the mesencephalon (MLd) and surrounding regions, we tested if MLd selectively responds to learned songs. In sequentially tutored birds, both right and left MLd were activated by the song stimuli (MLd$_{right}$: $F_{max} = 5.6$, $P_{uncorrected} = 0.001$; MLd$_{left}$: $F_{max} = 3.3$, $P_{uncorrected} = 0.03$). In contrast, the left MLd ($F_{max} = 6.76$, $P_{uncorrected} = 0.02$) was activated in response to songs in control birds. Post hoc $t$ test revealed that the right MLd of sequentially tutored birds was activated due to a significantly higher BOLD response evoked by TUT2 songs (Fig. 3k: MLd$_{right}$: TUT2 > TUT1: $t_{max} = 1.76$, $P_{uncorrected} = 0.04$, TUT2 > NOV: $t_{max} = 2.71$, $P_{uncorrected} = 0.006$ or $P_{FWE} = 0.03$; MLd$_{left}$: TUT2 > NOV: $t_{max} = 2.34$, $p_{FWE} = 0.094$). A BOLD response that differentiated TUT2 song from other songs was thus detected in the right MLd of sequentially tutored birds (Fig. 3k). In control birds, the left MLD was selective for tutor songs (Fig. 3j: TUT > NOV: $t_{max} = 1.95$, $P_{uncorrected} = 0.03$).

To investigate the hemispheric differences in tutor-song selective responses in the midbrain/hindbrain cluster, we used a different ROI approach. For this, we calculated the percent signal change in functionally defined ROIs; these ROIs were composed of an ensemble of voxels in which significant activation in response to TUT1 (Fig. 3f; TUT1 > NOV in control birds) and TUT2 song (Fig. 3a: TUT2 > TUT1 in sequentially tutored birds) was observed in the voxel-based group analysis at 90 dph. In control birds, significant differences in BOLD responses to TUT1 and NOV songs (TUT1 > NOV) in the left hemisphere emerged after re-exposure to TUT1 as the birds reached adulthood (Fig. 4a: 90 dph: left vs right, $t_{(12)} = 3.4$, $P = 0.005$; 55 dph: left vs right, $t_{(12)} = 1.26$, $P = 0.23$). In contrast, in the sequentially tutored birds, TUT2-selective responses also emerged with age but were localized in the right hemisphere (Fig. 4b: 90 dph, left vs right, $t_{(14)} = -4.2$, $P = 0.0007$; 55 dph, $t_{(14)} = 0.54913$, $P = 0.59$). This ROI analysis thus confirms that tutor-song selectivity is localized in the midbrain region at the end of the sensorimotor learning period, but the hemispheric differences depend on the learning experience.

**Neural selectivity for tutor song is related to the strength of song learning**. Because there was a large range of learning outcomes (ranging from 0 to 100% similarity with TUT2), we investigated if differences in neural activation in response to TUT2 song were related to the fidelity of song imitation. The sequentially tutored birds were divided into "good" and "poor" learners based on the % similarity score with the second tutor, which was quantified as the fraction of shared elements between the tutee- and the second-tutor song (number of syllables copied by the tutee from the total number of tutor syllables, multiplied by 100). Birds with scores higher than the median were assigned to the "good" learner group, and lower than the median to the "poor" learner group. When comparing good and poor learners at 90 dph, we found TUT2-selective responses in good learners but not in poor learners in the left NCL (Fig. 5b, e: NCL$_{left}$: $t_{max} = 3.27$, $P_{uncorrected} = 0.003$; NCL$_{right}$: $t_{max} = 3.1$, $P_{uncorrected} = 0.004$, cluster spans four voxels which does not meet criteria for significance). Before they had learned the second song, at 55 dph, good learners showed increased activation for TUT2 song relative to poor learners in the nidopallium (Fig. 5a: $t_{max} = 5.9$, $P_{FWE} = 0.08$ or $P_{uncorrected} < 0.0001$), which included but was not limited

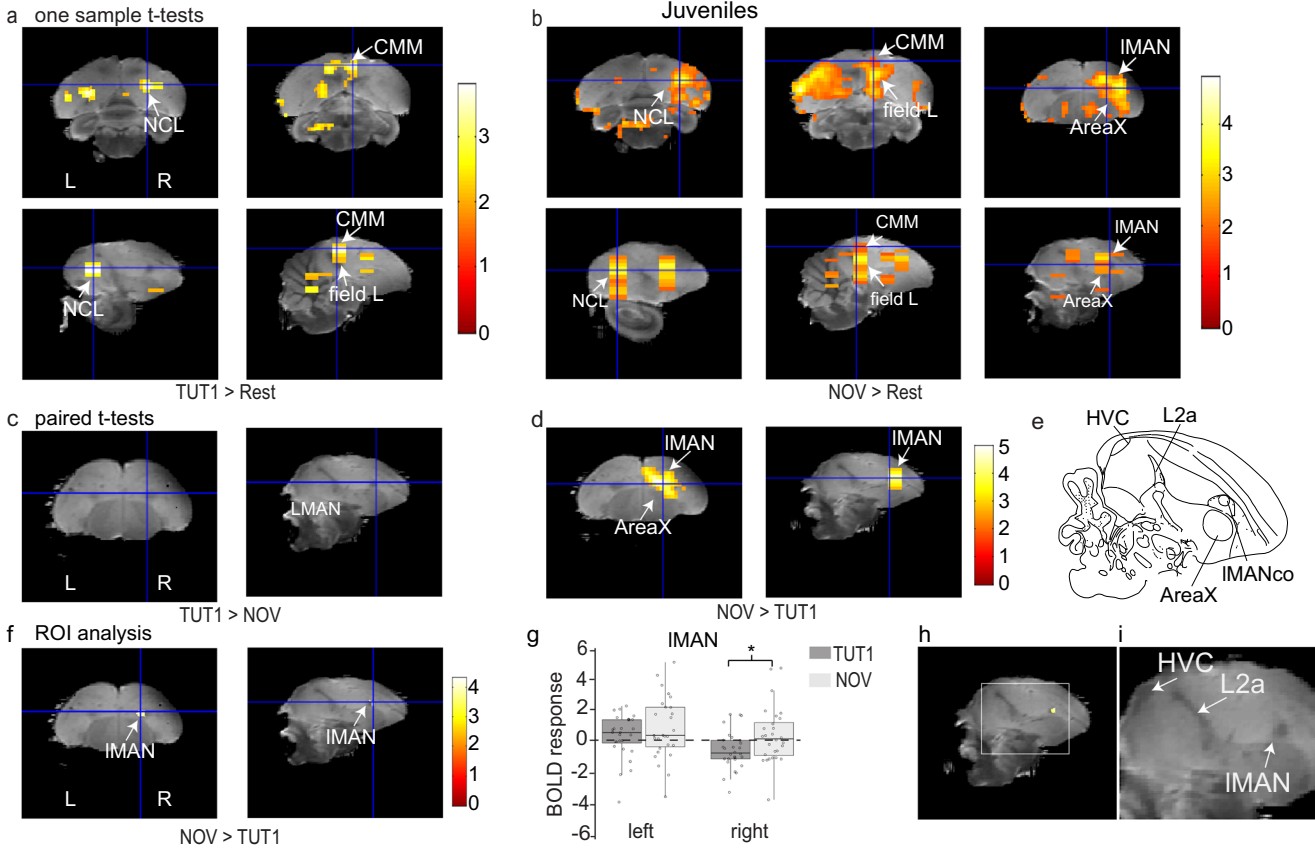

**Fig. 2 Brain regions activated by song in 55-day-old birds. a**, **b** Statistical map of voxels activated by song superimposed on coronal and corresponding (indicated with cross hair) parasagittal sections of the high-resolution atlas of the zebra finch brain[84]. Only voxels with t > 1.76 (one-sample t tests, exploratory threshold $P_{uncorrected} < 0.05$) are displayed (A: TUT1 > rest; B: NOV > rest); t-values are color-coded based on the scale on the right. **c**, **d** Paired t tests show a significantly higher BOLD response in forebrain regions, including lMAN, in response to novel song (NOV vs. TUT1 playback). All voxels with t > 2.47 are displayed (paired t tests, $P_{uncorrected} < 0.01$) and t-values are color-coded based on the scale displayed in the figure. **e** Line drawing of a parasagittal section from the zebra finch histological atlas[90]. **f** Region-of-interest (ROI) analysis restricted to lMAN shows a higher BOLD response to NOV vs. TUT1 in the right lMAN. **g** Boxplot of BOLD response (β weights) for each stimulus relative to rest periods within lMAN (indicated with a white arrow in (**f**)). Boxplots showing the interquartile range (box), median (black line), and 1st and 3rd quartile; each circle represents data from one individual bird (*$P_{FWE} < 0.05$, N = 28). For visual representation, one data point (bold estimate = −9.50416, left hemisphere, NOV) was excluded from the figure but not from the statistical analysis. **h**, **i** Detailed views of the exact location of the activated cluster observed within lMAN. NCL: caudolateral Nidopallium, CMM: caudomedial Mesopallium, lMANco: core of the lateral magnocellular nucleus of the anterior Nidopallium, L2a: subfield L2a of field L. HVC and Field L are used as proper names.

to NCLm (Fig. 5c; Von Eugen 2020), a small cluster in CMM (Supplementary Fig 2a: CMM: $t_{max} = 3.18$, $P_{uncorrected} = 0.004$) and another cluster in field L ($t_{max} = 3.91$, $P_{uncorrected} = 0.001$). Interestingly, after learning the second song, a positive correlation emerged between TUT2 selectivity (defined as the amplitude of TUT2 minus TUT1) and the amount of song learned from the second tutor (Fig. 5e, f: $NCL_{left}$: $r = 0.44$, $P = 0.003$, $NCL_{right}$: $r = -0.05$, $P = 0.63$). Thus, the more a bird has learned from its second-song tutor, the stronger the BOLD response to TUT2 (as compared to TUT1) in the NCL.

## Discussion

In social animals, such as humans and zebra finches, sensory experience in early development tunes receptive fields in the auditory cortex which results in neural selectivity for conspecific vocalizations[23–27]. In zebra finches, the NCM, a secondary auditory region, may even encode perceptual memories of the song from the tutor to which the bird was exposed early in life. In this region, expression of immediate early genes, rates of neurogenesis, and habituation of neural responses have all been shown to be related to the strength of song learning[20,28–33].

Pharmacological inhibition or lesioning of the NCM impairs song learning and reduces the behavioral preference for the tutor's song[34–36]. In this study, we used fMRI to understand memory-related neural changes in song processing due to sequential song learning in zebra finches. Zebra finches exposed to a second-song tutor exhibited learning-related changes in an auditory midbrain nucleus and in the caudal lateral Nidopallium (NCL), the avian analog of the mammalian PFC.

**Representation of song memory from the first tutor.** In parallel with the birds' behavioral flexibility to imitate syllables from two song tutors at different timepoints in development, we found evidence that the neural representations of early auditory memories and memories of recent auditory experiences became localized to different hemispheres, as compared to birds that were raised with a single song tutor. 55-day-old juveniles reared normally with their first tutor (TUT1) until the age of 32 dph did not show a TUT1-selective response (as compared to novel conspecific songs) despite the fact that the birds had behaviorally learned elements from their tutors' song (Fig. 1). Instead, they exhibited a higher BOLD response across the auditory lobule to

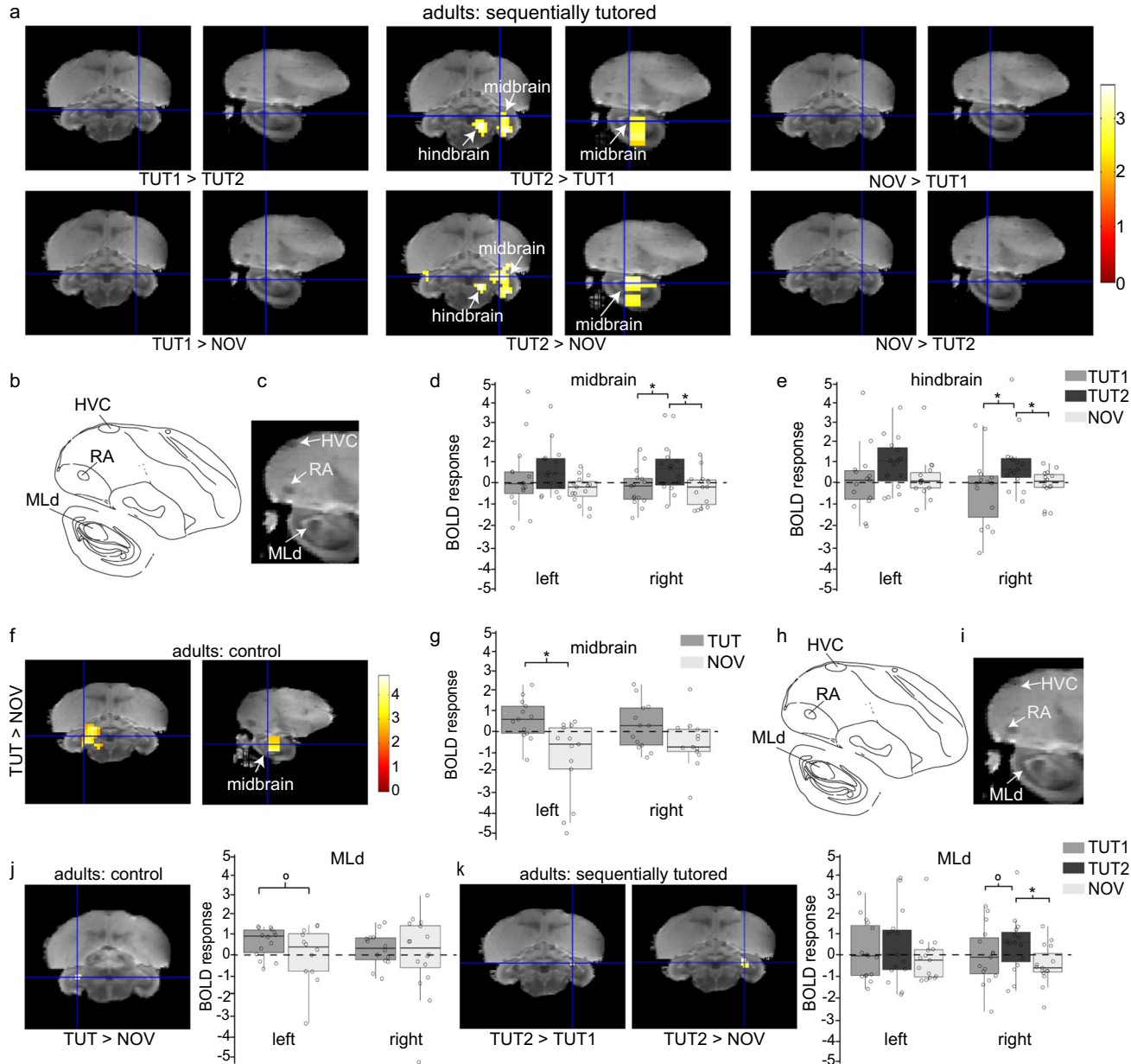

**Fig. 3 Brain regions activated by song in 90-day-old birds. a** Statistical maps of all post hoc $t$ tests performed within the main effect of stimulus in sequentially tutored birds. Blue cross-hairs indicate the location of MLd in the zebra finch MRI atlas. **b, c** Detailed view of the exact location of the activated cluster observed within the midbrain. **d, e** BOLD response (β weights) elicited by each stimulus relative to the rest periods in the clusters indicated with white arrows in (**b**). Boxplots showing the interquartile range (box), median (black line), and 1st and 3rd quartile; each individual bird is represented by a circle (*$P_{uncorrected}$ < 0.01, $N$ = 15). **f, g** Statistical map of BOLD activation induced by tutor over novel conspecific songs (paired $t$ tests) in control birds and the corresponding BOLD response (β weights) elicited by TUT and NOV song relative to rest periods. Boxplots showing the interquartile range (box), median (black line), and 1st and 3rd quartile; each individual bird is represented by a circle (*$P_{uncorrected}$ < 0.01, $N$ = 13). All voxels with $t$ > 2.46 ($P_{uncorrected}$ < 0.01) are displayed. **h, i** Detailed view of the exact location of MLd. **j, k** Region-of-interest (ROI) analysis restricted to MLd shows that control birds (**j**) have a stronger differential BOLD response in left MLd while sequentially tutored birds (**k**) have a stronger differential BOLD response in right MLd. $T$ values are color-coded according to the scale displayed on the right. $P_{FWE}$ < 0.05 considered as statistically significant for ROI analysis. MLd: dorsal lateral nucleus of the mesencephalon, RA: robust nucleus of the arcopallium. HVC is used as a proper name.

novel conspecific songs as well as a novel song selective BOLD response in lMAN. This is contradictory to previous electro-physiology studies: single neurons in juvenile lMAN were found to be selective for either the tutor or the bird's own song in normally reared zebra finches[16,37] or for bird's own song or the first tutor song (TUT1) in juveniles prior to exposure to a second-song tutor[16]. This difference could potentially be due to the anesthetics used in electrophysiology and fMRI studies (urethane

vs isoflurane) or to the difference between population activity (BOLD signal) and single-cell activity (electrophysiology). Nonetheless, lMAN is important for juvenile vocal learning[38,39] as inhibition of NMDA receptors during tutoring sessions in juvenile zebra finches impairs song copying, suggesting a role in sensory memory acquisition[40].

Even though a tutor-song-selective BOLD response was lacking in the juvenile brain, the higher BOLD response to novel

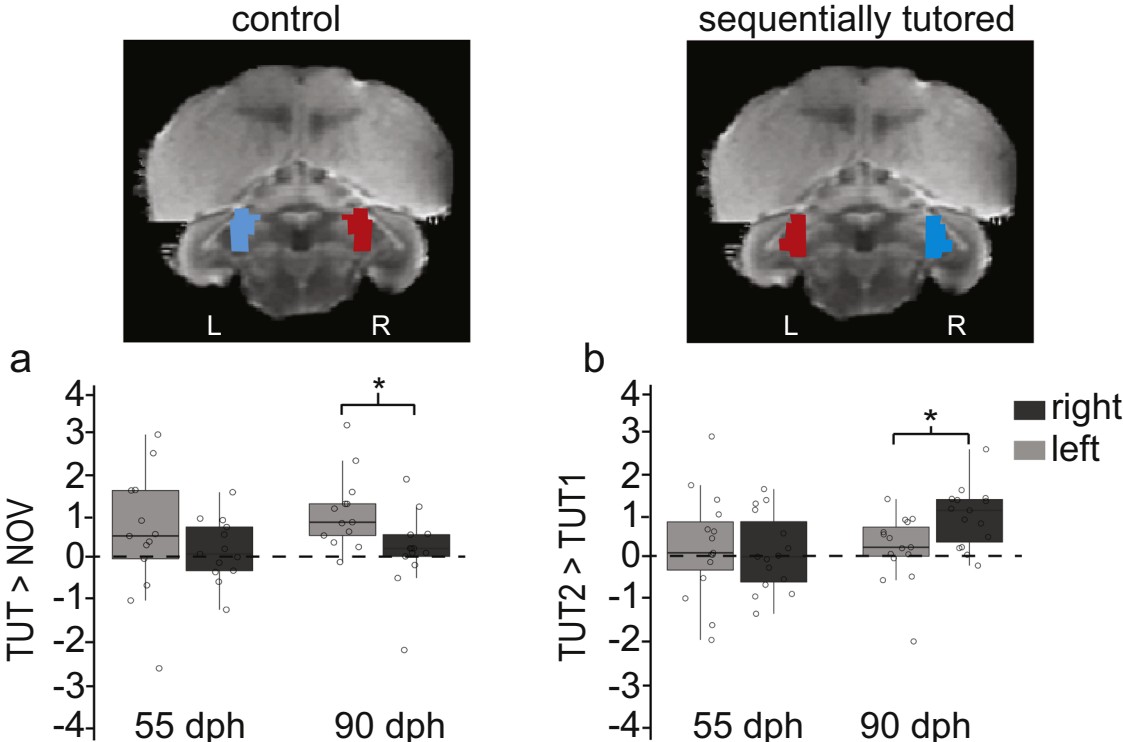

**Fig. 4 Development of lateralized tutor-selective responses.** Illustrations (top) of the activated cluster (blue) and mirrored (red) regions of interest for the lateralization analysis overlaid on coronal sections of the zebra finch MRI atlas. Estimates of differential processing (bottom) of TUT1 songs in (**a**) control birds (% signal change: TUT > NOV), and (**b**) TUT2 songs in sequentially tutored birds (% signal change: TUT2 > TUT1) in the left and right midbrain. The regions of interest for calculating percent signal change were based on the activated clusters for TUT > NOV in Fig. 3f and TUT2 > TUT1 in Fig. 3a. Boxplots showing the interquartile range (box), median (black line), and 1st and 3rd quartile; all birds are represented by circles (control: N = 13, sequentially tutored: N = 15). Paired two-tailed t tests were performed, *P < 0.05 between left and right hemispheres.

conspecific songs indicates that there are brain regions which differentiate between learned and novel songs, and thus a memory representation of tutor song must exist in the juvenile brain. This memory representation may either be sparse or dispersed, making it undetectable by fMRI. At the end of the sensorimotor learning period, TUT1 selectivity emerged in a small cluster in the left NCM, which did not meet the threshold (see Supplementary Fig 3). Albeit not reaching this threshold, the TUT1-related activity found in NCM could be biologically relevant as it is consistent with previous research, showing that NCM neurons are modulated by experience and are selective for learned vocalizations[30,41–43].

Another factor that could confound our results is the auditory isolation of birds in between tutoring sessions. However, it is unlikely that the higher BOLD response to NOV over TUT songs in juveniles is the result of acute isolation. If that was the case, we would also expect a higher BOLD response to NOV songs in adult zebra finches after a similar period of isolation, which we did not find. On the contrary, the social isolation between 32 and 55 dph may have extended the sensory acquisition period[16,44].

**Memory for the second-tutor song in the adult brain.** Birds that learned one song only (control group), and birds that learned the song from both their first and their second tutor (TUT2), showed TUT-selective responses for the song of the most recent tutor in the midbrain and hindbrain regions. Sensory experience with the tutor resulted in the development of tutor-selective responses in the auditory midbrain (MLd), but the hemisphere in which selectivity emerged at the end of the learning period was different depending on the specific learning experience (left for control birds and right for sequentially tutored birds). Although we

cannot rule out a contribution of age and brain maturation in the development of TUT-selective signals observed in the midbrain, the timing of tutoring periods and isolation periods were identical for birds in the control and sequentially tutored groups. This suggests that the emergence of TUT-selective signals reported here at the end of the learning period was the result of the specific tutor experience (single or sequential) during the sensorimotor learning phase.

Electrophysiological studies have shown that the presence of tutor-song-selective responses in the midbrain of adult zebra finches is dependent on early life experiences as well. For example, raising zebra finches with Bengalese finch foster parents can significantly alter sensory coding in the auditory midbrain and forebrain[45]. In addition, a different fMRI study has also shown that birds store representations of vocal memories in the auditory midbrain[46]. However, van der Kant et al.[46] reported selectivity for tutor song in the right auditory midbrain (MLd), similar to TUT2 selectivity found in sequentially tutored birds (this study). In this study, all birds, whether or not they were single or sequentially tutored, were normally reared with an adult male during the early sensory period (0-32 dph). The birds in van der Kant et al.[46] were separated from their father at 7 dph and exposed to a tutor at 43 dph. Thus, hemispheric differences in tutor-selective midbrain activity between our control single-tutored birds and those in the study by van der Kant et al.[46] could be a result of lack of tutor experience during the sensory period in the latter study (see also ref. [47] for effects on attentive listening behavior and song learning caused by tutor deprivation between 0 and 45 dph).

Hemispheric differences observed in MLd in Fig. 3g, e could also be attributed to differences between songs due to negative

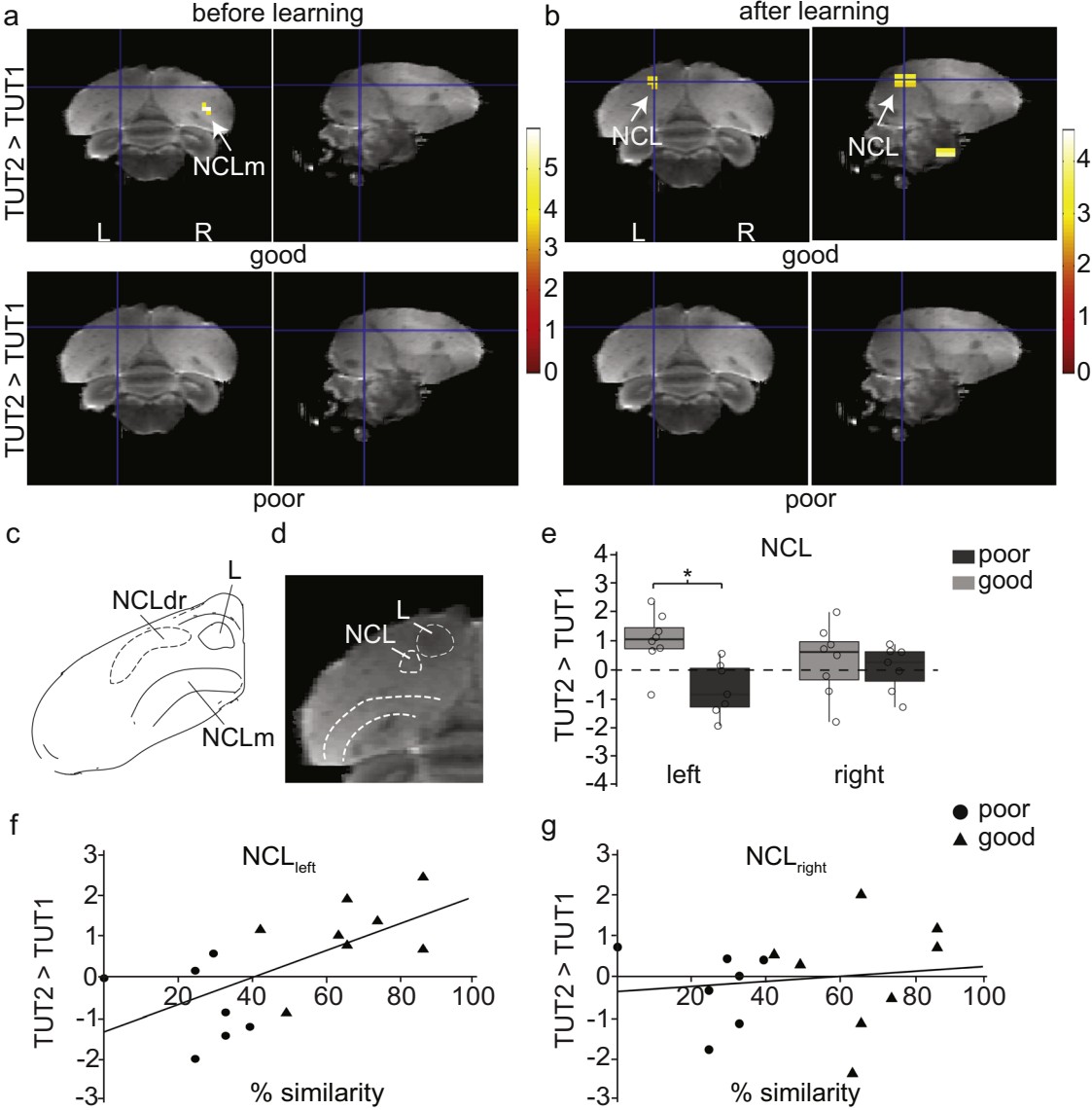

**Fig. 5 Good learners show a selective response in the NCL after learning the second-tutor's song, as compared to poor learners. a** Before exposure to the second-song tutor, there is already a TUT2-selective response (TUT2 > TUT1; all voxels with $t > 2.65$, $P_{uncorrected} < 0.01$ are displayed) in the nidopallium in juveniles which go on to learn well from the second tutor (good learners), as compared to those which do not (poor learners). **b** After exposure to the second-song tutor, there are TUT2-selective responses (TUT2 > TUT1; all voxels with $t > 2.65$, $P_{uncorrected} < 0.01$, are displayed) in part of field L (L2a) and part of the NCL in adults that successfully imitated TUT2 song (good learners) compared to poor learners. Statistical map of voxels superimposed on a high-resolution MRI atlas of the zebra finch brain. **c** Line drawing of a coronal section (A1.35) from the zebra finch stereotaxic atlas[89] showing the areas NCLdr (lateral to field L) and NCLm, based on tyrosine hydroxylase (TH) positive fiber distribution within the caudal nidopallium (adapted from refs. [22,89]). **d** Detailed view of the exact location of the cluster observed within the NCL of good learners in (**b**). **e** BOLD response (β weights) between different stimuli in the cluster indicated with a white arrow in 5b after learning the second-tutor song. Boxplots showing the interquartile range (box), median (black line), 1st and 3rd quartile, and all data points (*$P_{uncorrected} < 0.01$, $N = 15$). **f, g** After learning the second-tutor song, in adult birds, the strength of TUT2-selective signals in the left NCL (F, $NCL_{left}$: $r = 0.64$, $P = 0.01$), but not in the right NCL (G, $NCL_{right}$: $r = 0.13$, $P = 0.63$), was related to the fidelity of song imitation. The regression line is shown for good and poor learners. TUT2 selectivity is expressed as the mean amplitude estimate of differential BOLD signals of [TUT2 minus TUT1] in non-dimensional units. % Similarity is expressed as the percentage of shared elements between the tutee's song and the second-tutor's song. The median similarity score was used to divide birds into "good" learners (scores higher than the median) and "poor" learners (scores lower than the median). Black triangles represent birds with good imitation scores and black circles represent birds with poor imitation of the second-tutor song. NCL: caudolateral nidopallium, NCLdr: rostral aspect of the dorsal caudolateral nidopallium, NCLm: medial aspect of the caudolateral nidopallium, L: field L.

responses while listening to NOV songs. Positive BOLD responses reflect increased neural activity, but negative BOLD responses could either be because of decreased neural activity[48–51] or "vascular steal"[52]. The BOLD responses to NOV songs in MLd are characterized by large variability between birds and may not be robust enough to speculate about the underlying neural processes that could cause a negative BOLD response. However, it is interesting to note that both TUT1 and TUT2 show positive BOLD responses in the same region where NOV songs show no BOLD response or a negative BOLD response. This may represent neuroadaptive changes associated with learning, through which the TUT songs become more salient because of

familiarity, while at the same time the sensitivity towards NOV songs is reduced.

**The NCL is involved in behavioral flexibility during vocal learning.** Surprisingly, when differences in brain activation between TUT1 and TUT2 were compared between good and poor learners, successful learners of TUT2 exhibited activation in NCL, a region analogous to the prefrontal cortex in humans and adjacent to the secondary auditory region NCM. Interestingly, the greater the fidelity of song imitation, the higher the BOLD response in the left NCL, which suggests that the NCL may be involved in neural plasticity necessary to consolidate a sequential memory of tutor song. The avian NCL is a higher-order multi-modal forebrain region, densely innervated by dopaminergic fibers from the ventral tegmental area (VTA) and substantia nigra (SN) and highly interconnected with other sensory and motor regions[22,53–58]. The function of the caudocentral nidopallium (NCC) and lateral NC in zebra finches is starting to be elucidated through recent fMRI, IEG, and electrophysiological studies. The NCC seems to play a role in evaluating courtship signals in females[59]. Neural activity in males (expressed as IEG expression) varies in response to song type (tutor vs conspecific), behavioral state (singing vs non-singing), and developmental stage (juveniles vs adults): juveniles exposed to tutor song while they were singing themselves showed high ZENK induction in dorsal NCL, while this was not the case in adults or in juveniles that were only listening to tutor song[60]. Interestingly, a small set of neurons in NCL in pre-singing juvenile male zebra finches have been shown to be highly selective for tutor song[60,61]. Both of these studies suggest that the NCL is involved in song learning in males, and perhaps in evaluating the Bird's Own Song resemblance to the tutor's song, similar to NCC's role in the evaluation of courtship songs in females. Our study further shows that successful imitation of a second song is related to tutor-song-memory dependent neural activity in the NCL, and opens up new avenues to explore the role of NCL in song learning.

**Parallels with human second language learning: a potential animal model for phonetic aspects of bilingualism?** In humans, language centers are generally left-lateralized, but hemispheric dominance varies considerably among bilingual people who typically exhibit greater right hemispheric involvement in com-prehension tasks[62,63]. Monolingual people proficient in their native language show a leftward bias with proficiency and age[64–66]. In contrast, children and adults who are in the process of learning a second language show more bilateral or right-dominant activity in brain regions involved in speech perception[64,67]. When non-native language proficiency increases, reading and speech comprehension displays significant variability in hemispheric dominance, while verbal production remains left-lateralized in late language learners[68]. Similarly in zebra finches, perceptual responses are lateralized in birds reared with a single song tutor, with the right hemisphere showing higher BOLD responses for the tutor and the bird's own song in the NCM/field L region in one study[69], and for tutor, bird's own song, and conspecific songs in the MLd in two other studies[46,70]. Our fMRI data in zebra finches expands our understanding of hemispheric differences in auditory processing by comparing single-tutored and sequentially tutored birds. Perceptually-differentiating responses in birds that learned a second song were found in the opposite (right) hemisphere as compared to birds that learned only one song (left; this study). Bilingual people also recruit regions other than classic language areas for processing the sec-ond language[7,63,71]. At the onset of learning, bilingual people show additional brain activity in the prefrontal cortex (PFC) and

with increasing proficiency, there is reduced activity in the PFC (reviewed in refs. [8,72,73]). In birds, just like in humans, higher-order regions (NCM[21] and NCL, this study) in the avian cortex (pallium) are recruited when additional plasticity is needed to learn a second song late in the sensorimotor learning period.

Even though zebra finches can acquire new syllables from a second tutor during the sensorimotor learning period, they, unlike humans, cannot flexibly switch between the two songs in adulthood. While the neural mechanisms of using and producing a second language or second song could be quite different, our results indicate that the strikingly similar neural patterns we report for birds could be related to the additional neural plasticity necessary for auditory–vocal learning late in development. Thus, the parallels we revealed here between bilingual people and sequentially tutored zebra finches open new avenues for examining systems-level mechanisms of speech acquisition. They expand our knowledge of brain regions involved in the acquisition of multiple auditory representations that guide vocal learning, further developing the zebra finch as an animal model for the phonetic aspects of human bilingualism.

## Methods

**Animals and rearing protocol.** Thirty-two male zebra finches were raised from 25 breeding pairs in the animal facility at Wellesley College. Birds were reared with their mother, father, and siblings until 30–32 days post hatch (dph; mean ± SD: 31.6 ± 0.85) at which they were transferred to individual housing in acoustic iso-lating chambers. Birds remained isolated in these cages until 55 dph, when they were co-housed with either a second adult tutor (TUT2, not the biological father; sequentially tutored group) or the biological father (TUT1; control group) for the next 10 days. At 66 dph, tutors were removed from the cage and birds remained isolated until the end of the experiment at 150 dph (Fig. 1b). All birds underwent two fMRI sessions: at 55 dph, when birds had only been exposed to their TUT1, and later at 90 dph, after exposure to TUT2 or re-exposure to TUT1 (Fig. 1b). All birds were kept on a 16:8 h light:dark cycle and were provided with seeds, grit, and water ad libitum. Experimental procedures were in accordance with US law and approved by the Institutional Animal Care and Use Committee of Wellesley College (IACUC #1405,1702, 2004).

**Behavioral analysis.** Vocalizations of tutors and tutees were digitally recorded with directional microphones (Shure SM93, Shure Incorporated, Niles, IL, USA) using custom-written software. Due to technical issues, songs from two subjects in the control group were not recorded at 55 dph, and from one subject at 90 dph, and thus these birds were excluded from the similarity analysis. In order to select second tutors, we used asymmetric measurements in Sound Analysis Pro[74]. The automated software quantitatively measures percent similarity between two songs. The percent similarity score calculated by the software is a cumulative score based on five acoustic features of the song: entropy, pitch, frequency modulation, and spectral continuity[75]. Song files were selected from recordings when the tutors were housed alone in soundproof chambers. The most frequently repeated motif was identified and ten random examples were selected of this motif between 2 pm and 11 pm. Ten motifs of the first tutor (TUT1; $N = 14$) were compared to ten motifs of the prospective second tutor (TUT2; $N = 14$) to calculate the average percent similarity. After the song-similarity scores between the first tutor and prospective second tutors were determined, second tutors were chosen based on low similarity (<50%) between the first and second tutor. There was no significant difference between motif duration between the two tutors (TUT1: mean 1.03 ± SD 0.23 s; TUT2: mean 0.85 ± SD 0.18 s; two-sample-tests, $P > 0.05$). The number of syllables in TUT1 was 6 ± 0.7 and the number of syllables in TUT2 was 5 ± 1.2 (mean ± SD).

To determine the similarity between the tutee's song and its two tutors, we used a different approach. Because of the two-tutor rearing environment, birds often included syllables from both tutors in their crystallized songs, or complex syllables consisting of notes from both tutors, which was difficult to accurately assess with Sound Analysis Pro. After consultation with O. Tchernichovski, who is one of the designers of Sound Analysis Pro, we decided to use a panel of human observers to obtain a more accurate measurement of song imitation[16,29,76–78]. Two human observers (blind to whether a motif was from the first or second-song tutor) were instructed to assess the similarity between all syllables from two spectrograms (see Supplementary Fig 4 for examples of spectrograms). A score between 0–3 was given for every syllable of the pupil's song (0 being the lowest resemblance to a specific tutor syllable and 3 being the highest). We calculated percent similarity between two songs by dividing the number of syllables copied from a given tutor (defined as those with a score higher than 1) by the total number of syllables in the tutor song and multiplied by 100[16,29,76]. Tutee motifs were also compared with motifs from novel conspecific birds (used as 'NOV' stimuli in the fMRI experiments) to determine the baseline level of similarity between two zebra finch

songs. Based on the similarity score with TUT2 song in 90-day old birds, we divided the birds into two groups: "good" and "poor" learners. Birds with scores lower than the median of the group were categorized as "poor" learners and birds with scores higher than the median learning score as "good" learners[33,79].

**Data acquisition.** fMRI data were collected on a vertical 9.4 Tesla 400 MHz NMR system (Bruker Avance, Germany) using Paravision 4.0 software following a customized protocol. Briefly, birds were anesthetized with isoflurane (induction: 2%; maintenance: 1.5%) in oxygen (99.5%) at a flow rate of 400 cm³/min using a VIP Veterinary Vaporizer (Colonial Medical Supply Co., Franconia, NH). Anesthesia was delivered through a beak mask embedded in a customized bird holder for MRI scanning (see Supplementary Fig. 4d for pictures of the experimental setup). Throughout the experiment, the temperature inside the magnetic bore was maintained at 36 °C to facilitate regulation of body temperature. Bird's respirations were measured using a pressure transducer pad compatible with BioTrig Builder 1.01 Software and monitored visually.

We first evaluated gradient-echo and EPI sequences, but these resulted in image distortion due to the high magnetic (9.4 T) field. Spin-echo T2-weighted contrast prevented artefacts at high field MR resulting from brain-air interfaces, which are more prevalent in birds than mammals[80]. Thus, a 2D multi-slice rapid acquisition relaxation-enhanced (RARE) anatomical scan of high resolution (TE$_{eff}$/TR: 60/3000 ms, interslice gap/slice thickness: 0.75 mm, spatial resolution: $0.097 \times 0.097$ mm², acquisition matrix: $256 \times 256$ voxels; FOV: $25 \times 25$ mm) was acquired for each bird in the same orientation as the functional scan to facilitate spatial registration. For fMRI, a time series of T2-weighted RARE volumes consisting of 15 slices covering the whole brain was acquired with the following parameters: TE$_{eff}$/TR: 60/2000 ms; acquisition matrix: $64 \times 64$ voxels; slice number: 15; spatial resolution: $0.39 \times 0.39$ mm²; interslice gap: 0.75 mm; orientation: axial; RARE factor: 8; FOV: $25 \times 25$ mm; fMRI scan duration: 80 min (55 dph) or 120 min (90 dph). Because T2 relaxation time for venous blood is less than 10 ms at 9.4 T, the relatively long TE associated with the SE pulse sequences allows for sufficient spin dephasing in large blood vessels, thereby removing their contribution to the BOLD signal, which can be largely attributed to the extravascular BOLD signal from the small capillaries responding to an increased metabolic demand[80,81].

**Auditory stimulus presentation.** During fMRI, animals were exposed to three kinds of auditory stimuli: songs from the first tutor (TUT1), songs from the second tutor (TUT2), and novel conspecific songs (NOV). Tutor songs were obtained from prior recordings of the adult males before tutoring sessions. Conspecific songs were songs from birds that the experimental birds had never heard before. Stimuli for the ON-OFF fMRI block paradigm were generated using PRAAT[82]. Each set consisted of either two or three stimulus types: TUT1, TUT2, and NOV; 5–6 different sets of song with 1–2 sec of silence in between were concatenated to make a 32-second-long stimulus[83]. The length of these inter-song intervals resembles the rest intervals found in bouts of the undirected song. The stimuli were equalized across motifs for root mean square amplitudes in PRAAT.

Auditory stimuli were presented in a fixed order in an ON/OFF block design consisting of 32 s of auditory stimulation (ON) followed by 32 s of silence (OFF). Two additional dummy scans were added after every OFF block but were not used for analysis. An fMRI session for a 55-day-old bird consisted of 50 ON blocks (25 per stimulus, TUT1 | NOV) and 50 OFF blocks, and an fMRI session at 90 dph consisted of 75 ON and 75 OFF blocks playing three stimuli (TUT1 | TUT2 | NOV) in the same order. Two T2-weighted RARE images were acquired during each block, resulting in 50 images per stimulus type, per subject at 55 dph and at 90 dph. Auditory stimuli were cued by NBS Presentation software (Version 18.0, Neurobehavioral Systems, Inc., Berkeley, CA; www.neurobs.com) and delivered to the birds through a custom-designed speaker (20 mm 8 Ω Digi-Key #102-1542-ND magnets removed) embedded centrally in the bird holder.

**MRI data processing.** The following inclusion criteria were applied to the fMRI time series: (1) limited head motion (<0.5 mm translation or rotation in any of the three directions as measured from the realignment read out of the SPM realignment step), and (2) detection of a positive BOLD response to any of the auditory stimuli in the auditory regions[46,59,84,85]. In the event any one of the criteria was not met, subjects were discarded from the analysis. Out of 32 animals subjected to fMR imaging, four were excluded. One subject had head motion >0.5 mm and was removed from the analysis. Two subjects had no BOLD activation in both imaging sessions and were excluded from the analysis. One subject responded to auditory stimuli in only one of the sessions and was excluded from the repeated measures ANOVA as well as from that individual session. The success rate of detecting auditory stimulation in the current study was comparable with previous fMRI experiments in zebra finches[59,85]. Statistical Parametric Mapping (SPM12, Wellcome Trust Centre for Neuroimaging, London, UK; http://www.fil.ion.ucl.ac.uk/spm) was used for post-processing and voxel-based statistical analysis of fMRI data. fMRI time-series scans from each subject were realigned to correct for intra-individual head motion using a six-parameter rigid body spatial transformation (without re-slicing) in SPM12. fMRI scans were masked to exclude non-brain tissue, and a Gaussian Kernel of 0.5 mm full width at half maximum (FWHM) was

applied to achieve in-plane smoothing using custom-written scripts in MATLAB (R2015b). The realigned images were then co-registered to their own 2D multi-slice RARE anatomical scan using affine registration with the FLIRT tool[86] in the data processing software "FMRIB Software Library" (FSL, https://fsl.fmrib.ox.ac.uk/fsl/). In parallel, the 2D multi-slice RARE anatomical scan of each subject was spatially normalized with the high-resolution zebra finch MRI atlas[84] with ANTs using mutual information as a similarity metric (http://stnava.github.io/ANTs/). The transformation matrix generated in the previous step was then applied to the realigned and co-registered functional data using ANTs apply transform (http://stnava.github.io/ANTs/), resulting in functional data co-registered to the atlas. The down-sampled atlas was used as a reference point in the previous step for warping functional images to the atlas space which kept the functional image resolution at $64 \times 64 \times 16$. A high pass filter of 320 s was applied to filter out the low-frequency drifts in the BOLD signal. For first-level analysis for each subject, the BOLD response was modeled as a box-car function convolved with a hemodynamic response function within a general linear model framework. The first-level design matrix was created for each subject imaged at 55 and 90 dph, with auditory stimuli as "Conditions" and the six estimated movement parameters derived from the realignment corrections included as regressors in the model to account for residual head movement. After estimating GLM parameters (β), t-contrast images (containing weighted parameter estimates) were computed for different comparisons of auditory stimuli vs. Rest (TUT1, TUT2 and NOV) and between the different auditory stimuli (TUT1 > TUT2, TUT2 > TUT1, TUT1 > NOV, TUT2 > NOV, NOV > TUT1, NOV > TUT2). These t-contrast images were then taken to the second level (group) analysis. All images were acquired with the same background noise coming from the magnet and radio frequencies. Thus, these t-contrast images represent the additional BOLD activation due to the auditory stimulation, as any two images (e.g., Stimulus vs. Rest or Stimulus 1 vs Stimulus 2) were acquired in the same noisy environment.

**Statistics and reproducibility.** Wilcoxon signed-rank tests were used to determine if the birds had learned from their first or second tutor. An inter-rater reliability analysis using Kappa statistics was performed to determine consistency among the human observers' song scores, which showed that the scores assigned were reliable (Kappa = 0.67, $P < 0.0001$). $P$ values less than 0.05 were considered statistically significant. All statistical analyses were performed in R version 3.5.1.

Voxel-based group analysis of the fMRI data was performed in SPM12 (Wellcome Trust Centre for Neuroimaging, London, UK; http://www.fil.ion.ucl.ac.uk/spm). The Family Wise Error (FWE) method was used to adjust $P$ values to the number of independent tests performed. FWE corrections use the Random Field Theory to calculate the number of independent tests, considering the number of voxels as well as the amount of autocorrelation among the data[87,88]. For whole-brain analysis, FWE correction appeared too conservative to detect any effect (which is common in small animal fMRI, see[59,85]); therefore, analysis at the whole-brain level was done without correction for multiple comparisons ($P_{uncorrected}$) at a restrictive threshold of $P < 0.01$ with a minimum cluster size of five voxels. Results are reported by the highest voxel $T$ value within each cluster ($t_{max}$) with its associated $P$-value. To assess changes in brain activation due to tutoring experience, a repeated measures analysis was performed with the two fMRI sessions (session 1: 55 dph, session 2: 90 dph), and interaction between stimulus and age was determined. Single-subject t-contrast images (TUT1 > Rest, TUT2 > Rest, NOV > Rest) were used for the group level analysis in a flexible factorial design with subjects as random variable (2 × 3 repeated measures design; 1st within-subject factor: age at which fMRI sessions were performed [55 and 90 dph], 2nd within-subject factor: song stimulus [TUT1, TUT2, and NOV]. A whole-brain analysis was performed, followed by post hoc $t$ tests (one-tailed) only on the voxels that showed a significant interaction between song stimulus and age to compare responses to stimuli during each tutoring experience.

To determine the brain regions selective for TUT1 and NOV at 55 dph, after exposure to the first tutor only, single-subject t-contrast images (TUT1 > Rest and NOV > Rest) were entered in a paired $t$ test design, (TUT1 > NOV and NOV > TUT1) to determine brain regions selective for TUT1 and NOV ($P_{uncorrected} < 0.01$).

To determine the brain regions selective for TUT1 and TUT2 in 90-day old birds, single-subject t-contrast images (TUT1 > Rest, TUT2 > Rest and NOV > Rest) from the first-level analysis were entered in a One-Way ANOVA (within-subject factor: stimulus class with three levels). The main effect of the stimulus class was explored at the level of the whole brain ($P_{uncorrected} < 0.01$) and followed by one-tailed post hoc $t$ tests ($P_{uncorrected} < 0.01$) to determine brain regions selectively responding to TUT1 and TUT2 songs.

In order to compare activation for tutor and conspecific (novel) song between control ($N = 13$) and sequentially tutored birds ($N = 15$), two-sample $t$ tests were performed using TUT1 > Rest and NOV > Rest (first-level contrast images from each group). To determine TUT1-selective regions in birds reared with one tutor, a paired $t$ test was performed using TUT1 > Rest and NOV > Rest images from the first level. A two-sample $t$ test ($P_{uncorrected} < 0.01$) was performed to compare differences in TUT2-selective responses between good and poor learners before (55 dph) and after learning TUT2 song (90 dph). A functionally defined ROI was created from the activated clusters that were found with the two-sample $t$ test between good and poor learners in 55 and 90-day-old birds. A linear regression

analysis was performed for % similarity score between tutee and TUT2 song, and mean fMRI signal averaged over contiguous voxels in the ROI ($N = 8$ good learners and $N = 7$ poor learners, $P < 0.05$ considered statistically significant).

**Region-of-interest analysis**. A region-of-interest (ROI)-based analysis was performed with functionally defined regions of interest in each hemisphere based on the midbrain and hindbrain cluster that was significantly activated in the group analysis in 90-day-old birds. In order to compare responses between the hemispheres in control and sequentially tutored birds, a differential effect (% signal change) between TUT2 and TUT1 in sequentially tutored birds, and TUT and NOV in control birds was calculated for each subject imaged at 55 and 90 dph using Marsbar (http://marsbar.sourceforge.net/). The ROIs identified in one hemisphere were mirrored over the midline to obtain an identical ROI in the opposite hemisphere. Percent signal change over the contiguous voxels in each ROI was then compared across hemispheres using two-tailed paired $t$ tests. The results were considered significant at $P < 0.05$. The activated midbrain/hindbrain cluster observed in both control and sequentially tutored birds included MLd, the substantia nigra pars compacta (SNc), the ventral tegmental area (VTA), and the periaqueductal gray (PAG), but these brain regions, except for MLd, could not be delineated in the MRI atlas because of the lack of a differential contrast in MR images. Therefore, ROI analysis of specific nuclei within the larger ROI cluster only included MLd, which was identified and delineated using the zebra finch MRI atlas[84]. Although outside of the activated cluster, ROI analysis was also performed for the thalamic nuclei (DLM/Ov), but no significant results were found for control or sequentially tutored birds. In 55-day-old juveniles, the forebrain cluster that was significantly activated included lMAN and Area X, which were delineated using the zebra finch MRI atlas and paired $t$ tests were performed within those ROIs. Statistical differences between stimulus-evoked BOLD responses were assessed in MLd using one-way ANOVA in sequentially tutored birds, and paired $t$ tests in control birds. $P$ values were corrected for multiple tests using the Family Wise Error method and Random Field theory[88]. All region-of-interest analyses were performed on up-sampled functional images (dimensions:$128 \times 128 \times 128$).

**Reporting summary**. Further information on research design is available in the Nature Portfolio Reporting Summary linked to this article.

## Data availability
The datasets generated during and/or analyzed during this study are available from the corresponding author upon request. Supplementary Data 1 contains the source data behind the graphs.

## Code availability
Custom Matlab scripts that were used for in-plane smoothing of functional images using SPM functions are available from the corresponding author upon request.

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

## Acknowledgements

We thank Cassandra Pattanayak for advice on statistical analysis, Lynandrea Mejia '20 for help with image data collection, Taylor Burke '20, Lily Sall '21, Katie Barnes '21, Yasmin Sharbaf '21, Colleen Boggs '23, and Adrianne Cheung '24 for song data analysis, and Pat Carey, Nancy Thompson, and Valerie LePage for animal care. We thank all the

undergraduate research assistants who were involved in pilot experiments for the current study: Rachel Parker '13, who set up anatomical MR, Sarah Zemlock '14, who designed the speaker for inside the magnet, and Linlin Chen '17, Rebecca Jennings '17, Kethu Manokaran '18, and Rebecca Leu '18 who acquired preliminary fMR images. This work was supported by the Dr. Carol Angle Fund for Faculty Research, a Neuroscience Department Summer Research Fellowship, a Brachman-Hoffman fellowship, and by award R15HD085143 from the Eunice Kennedy Shriver National Institute of Child Health and Human Development of the National Institutes of Health. The publication fees for this article were in part supported by the Wellesley College Library and Technology Services Open Access Fund.

## Author contributions

S.M.H.G. conceived and designed the study, P.A. performed the study and developed the post-acquisition image analysis pipeline, P.A. and S.M.H.G. analyzed the data, S.P. designed the Arduino stimulus presentation and fMR image acquisition paradigm, P.P.K. contributed analysis tools, N.H.K. designed the MR and initial fMR image acquisition paradigm and provided technical expertise, and P.A. and S.M.H.G. wrote the paper. All authors have approved the final article.

## Competing interests

The authors declare no competing interests.
