## [Peer Review File · Communications Biology]

Reviewers' comments:

Reviewer #1 (Remarks to the Author):

The study of the team of Sharon Gobes found that acquisition of a second song (TUT2) changes lateralization of the auditory midbrain and that activity in the caudolateral Nidopallium (NCL), a region adjacent to the secondary auditory cortex, was related to the fidelity of second-song imitation. They demonstrated that during sensorimotor development, LMAN had stronger activation upon novel song exposure as compared to TUT1.

This is very nice work and very exciting data on zebra finches with clear relevance for human speech and second language learning.

These exciting findings however require some additional validation of the structures assigned to the data. To clarify: They claim involvement of LMAN, In figure 2, the activated region is huge using uncorrected data and one of these figures even shows Area X activation. To properly confirm this finding, one should show images of FWE corrected data in which the LMAN region should 'survive' and accordingly to priorly set criteria (eg if possible and as defined by literature: minimal cluster extend of 5 voxels). The data presentation would benefit from showing the outcome with cross hairs in 3 different orientations (or by even adding stereotactic coordinates eg according to Nixdorf or other). The same is true for fig 3 assigning NCM and field L which would also benefit from sagittal images with cross hairs. The same for midbrain, where MLd data are finally extracted ROI wise, where the corrected data should confirm the location of MLd. However LMAN and Field L, are still an easy target, as they are distinguishable with MRI contrast. The biggest hurdle however and the most interesting conclusion at the same time is the role of NCL which requires an accurate location of this structure (eg figure 5). The authors refer to the correct publication to verify this more carefully (von Eugen et al). Fig 9 in this publication shows the stereotactic coordinates and the section window (could be compared with yours) according to the Nixdorf atlas. The summarizing figure 12 in this publication illustrates how zebra finches do not have a single NCL structure but this structure is subdivided and spread over NCLd, NCLv and NCLm. Are you sure the activation is in an NCL subregion and not in NCC? Etc...Also here, displaying 'corrected' data in three different directions is mandatory and could lead to clear definition of NCL parts. Another remark on how the data are displayed is that when comparing activations upon different stimuli, one needs to display the same section(s) of the brain for each stimulus. In figure 3B for example, the bottom left picture has a different location and we cannot conclude on whether there were activations on this slice for the other stimuli. Drawing conclusions from the fMRI on specific regions is key: There is only one figure (fig 1) demonstrating location of regions and they are not mentioning the one referred to in the results (NCM, NCL, LMAN) and other locations than the one in figure 1 are used (fig 3B,H and 5B bottom left and fig 2 for LMAN). You should clearly overlay with atlas drawings in results section what are the regions you assign. In conclusion without stereotactic coordinates or contrast landmarks in the images, or 3D localization ...one cannot assign a specific brain structure/region/nucleus to the activated regions. I think the authors can adjust this.

Further remarks on results section:

Fig 1, legend: for consistency one should mention that red lines deal with NOV instead of unfamiliar

93: in the legend of Fig 1: red horizontal lines indicate mean similarity with unfamiliar birds. This is confusing as in general and also in the figures NOV is used? Or did I miss something?

168:PFWE<0.1 : is not significant !

Figure 5: the color coding is a bit too dark for good learners as compared to the color used in the figure

Remarks on the material and methods section:

482: How is the animal positioned and fixed in a vertical bore system ? head up or down? a figure could help.

482: 400 MHz instead of Hz

484: "Briefly" 1.5 – 2 % isoflurane. Authors do not specify whether this is initiation dose or also the maintenance level.

484: Two different flow rates are mentioned for isoflurane in oxygen. Specify the reason for the different flowrates and when each rate is used. Provide info on gas mixture % oxygen, % nitrogen or did one use pure oxygen? If so why? And are there publications doing this ?

487: "the temperature inside the magnetic bore was maintained at 36°C " did the authors determine the animal's temperature? Birds have 39 to 40°C body temperature. Both temperature and breathing rate are important indicators of animal physiology and influence the BOLD response (affinity of Hb for O₂).

492: 3D rapid acquisition: Is it 3D? looks like multi-slice 2D acquisition since you mention 0.75mm interslice gap ? what is exactly interslice gap? Does it comprise slice thickness? Seems quite large in a high-resolution acquisition. The FOV is not mentioned

497: Why did the authors choose such a high effective TE of 60ms ? What effect does it have on the images ? the paper does not show any fMRI data. Can the authors elaborate on which echo (of the 8) is used to calculate the effective TE?

499: Authors mention a scan duration of 120 minutes, is this total scan or fMRI scan duration. Under the paragraph "auditory stimulus presentation", the authors discuss two separate acquisition schemes for 55dph and 90dph, which consist of two block designs of different sizes.

511: The authors give more detailed information on their block design stimulation scheme. However, when calculating the scan time, this does not match with what has been mentioned in line

499 (120 minutes). 55dph; 300 repetitions = 80min, 90 dph; 450 repetitions = 120 min

512: What is the reason for the 1:2 ratio for ON:OFF blocks? Can the authors motivate this decision with literature or have additional experiments been performed to assess its benefits? Reducing the OFF-block duration would result in a dramatic decrease in acquisition time which is cost beneficial and reduces the anesthesia time for the animal.

514: Playing 3 stimuli in the same order... Normally this is randomized to reduce expectations/prediction and habituation to the stimulus. Why did one not choose to do this?

516; resulting in 50 images per stimulus type per subject: for which age group? Explain this number?

526: How was head motion assessed ? From realignment read-out or visually assessed before realignment? Was only translation or also rotation considered?

540: Add matlab version

536 – 547: Different transformation steps are mentioned. Were these transformations performed separately with different reslicing steps? If so, why weren't the separate transformations combined to prevent the (potential) additive bias of consecutive resampling steps?

547: "Precisely co-registered to the atlas" The authors do not specify the final resolution of the functional MRI data. Using the terminology "precisely co-registered" could be interpreted as an upsampling step to the high-resolution zebra finch MRI atlas.

558: all images 'where' acquired > should be 'were'

574: Whole brain voxel-based analysis is performed without family-wise error control. While the authors claim that this is too conservative for small animal fMRI, the results shown in the paper illustrate clusters that survive FWE correction.

Why is the very liberal threshold of $P_{uncorrected} < 0.05$ used instead of a more restrictive $P_{uncorrected} < 0.001$ or smaller which is better to control for false positive results?

Uncorrected data can be used exploratory and corrected data listed in table (eg peak voxel intensity and number of voxels). When claiming involvement of (new) regions one should test whether they survive correction

627: could not be delineated in the MRI atlas due to their size. It is more because they did not have a differential contrast in the images. Ov is indeed too small. Overall one could co-register with an atlas though....

Remarks on discussion section:

312: Another factor that could confound our results is the auditory isolation of birds in between tutoring sessions. Then an explanation follows which should overrule the potential effect of isolation. This explanation is not clear to me. Also the isolation between 32dph and 55dph, is there any literature information on whether this prolongs the sensory period? The sensitive/critical period? keeping some brain structures/nuclei sensitive for neuroplasticity for a prolonged period? and influencing the impact of TUT2? Why did you actually choose for this specific experimental setup with acoustic isolation periods in between? All this deserves a place in the discussion.

Finally I would change the title in the direction of: Tracing development of song memory with fMRI in zebra finches after a second tutoring experience. Maybe mention also the sensorimotor stage? I leave it up to the authors.

Reviewer #2 (Remarks to the Author):

In this study, the authors show that fMRI-based neural activity is associated with tutor song memory when juvenile zebra finches were sequentially tutored by two adults during the sensitive period of song learning. Behaviorally, juvenile finches were able to copy the second tutor song after the song memory acquired earlier from the first tutor. Some of the neuroimaging results are really interesting, the tutor song memory is identified in NCM and NCL, and the changes of lateralization in midbrain after the second tutor presentation. This study provides important findings about the neural basis of song memory acquisition during the sensitive period of sensorimotor learning. This manuscript is clearly written, and the statistics are valid. However, a few major issues need to be clarified.

1. For a fMRI study like this, a well characterized and quantitative behavioral measure is crucial. The authors should provide a more detailed description and examples of (1) selection and quantification of song similarity between tutors; (2) quantitative measure of percent similarity between tutor and tutee.

Quantitative measure of percent similarity between the first and second tutors (and a novel conspecific song). How many adults (sample size) were used as the first tutor and as the second tutor? If the second tutors were chosen based on low similarity between the first and second tutors, what is the "low" similarity quantitatively? Do the first and second tutors have similar duration of motif (or the same number of syllables per motif)? What specific function of similarity match was used in SAP for this quantification (symmetric or asymmetric scoring)? Can authors provide sonogram examples of the first and second tutor songs (and novel songs) in Figure 1 or a supplementary figure?

Quantitative measure of percent similarity between a tutee and its first or second tutor. I agree that it is a good idea to use two blind judges to observe and identify the song similarity score between a juvenile and its first or second tutor. To better visualize the similarity measure, it is a good idea to provide sonogram examples of a good similarity match to the first tutor vs. the

second tutor (as well as novel songs). Was sequential order of syllables (or syntax structure) used for this quantification?

2. Social isolation. In this study, juvenile finches were socially isolated from 32-55 days of age (more than 3 weeks of social isolation?) then again from 66-150 dph. Why placed the juveniles in isolation for so long? I guess the 3-week social isolation (and then provided a second tutor at 55 dph) probably could motivate isolated juveniles to interact with the second tutor and the juveniles are more likely to learn from the second tutor? Social isolation is also known to delay song development in zebra finches (Morrison, 1993), therefore this isolation may extend neural plasticity for song acquisition from the second tutor? What happened if a juvenile was exposed to the first tutor then to the second tutor without social isolation? It would be great to have a non-isolated group as a control.

As the authors noted, zebra finches are social and colonial birds. I understand social isolation is a relatively common experimental procedure for neuroethological study of birdsong learning, but several weeks of isolation seems quite dramatic for a social animal, especially the isolation (32-55 dph) was performed at the peak of the sensitive period for song learning in zebra finches. Could such "unnatural" social isolation delay or affect the developing auditory system, hemispheric differences in tutor-selective midbrain activity, or LMAN selectivity response (as previous electrophysiology or neuroimaging studies showed different results) during the sensitive period of song learning? For the future study, visual isolation or alternative methods (housing with a juvenile female bird, for example) might be more appropriate.

Minor issues:

1. Figure 1c. The behavioral results are quite interesting, I wish the authors can elaborate more of the behavioral results in this figure or in a supplementary figure. They can help us to better understand the neuroimaging results. What does each dot represent? It is interesting that some of the juveniles had a high similarity match (almost 100%) to TUT1 at 55 dph. Did this bird match well to TUT1 at 90 dph, or it modified its song syllables and syllable order (syntax) and matched TUT2? Was a good learner for TUT1 also a good learner for TUT2? Or does good learning from the first tutor impose constraints to learn from the second tutor? It would be nice to visualize the song imitation by showing sonogram examples of crystallized songs with high and low similarity matches.

2. Line 436, 32 male finches are from how many clutches?

3. Line 457, song motifs were selected between 2pm-11pm. 11pm seems to be really late for a diurnal species. What was the onset of the (16:8) light cycle? What was the reason to have a light cycle like this?

4. Line 504, Conspecific songs or Novel conspecific songs?

5. Line 507. 5-6 bouts of song recordings were from the repetition of the same song bout or 5-6 different song bouts (control of pseudoreplication)?

6. Discussion: Parallel with human second language. The authors discussed the parallel between birdsong learning from the second tutor and human second language learning. Although the comparison of hemispheric dominance between birdsong and language learning is interesting, there are limitations for this comparison. Zebra finches crystallize only one song. They are genetically "monolingual" and are not able to acquire, process, and produce two different song types sequentially and simultaneously. Male finches have the learning plasticity to modify their previously memorized song from the first tutor to match the later exposed second tutor during the sensitive period and end up crystallizing one song type for the lifetime. Also, the animals in this study were under "unnatural" isolation manipulation. The underlying neural mechanism could be very different from sensorimotor processing of second language learning.

Reviewer #3 (Remarks to the Author):

This is an interesting paper addressing the neural substrate for second tutor song memory, as a parallel for second language learning. I have some comments on and questions about the presentation and interpretation of the findings, but I think the authors should be able to address these comments without too much additional work.

General comments:

1) Can you please add a figure with song examples of both groups on 55 and 90 dph? I would like to get a better sense of the level of song imitation.

2) The regions with significant BOLD response are often much bigger than the anatomical regions that you subscribe the activation to (e.g., LMAN or MLd). Why do you think this is? Is the response not specific to the song nuclei? Is there a lot of anatomical variation between the birds? Something else?

3) Do you want to add an explanation about what neural processes are thought to be reflected by positive vs. negative BOLD responses? Some of your differences are caused by more negative BOLD responses (rather than by higher positive BOLD responses). For example, I think the poor learners in figure 5 have more negative responses in D than C, haven't they? Or 3G: right TUT2 is significant compared to slightly negative NOV and TUT1 responses; while in the left MLd there's a similar pattern but NOV and TUT1 are not negative and TUT2 is not significantly different.

4) It seems tricky to present the data as you do in figure 4, because now it may be tempting to interpret the midbrain lateralization result as a tutor-specific effect. However, it does not seem to be caused by a different response to tutor song in the two hemispheres, but by a different response to novel: In figure 3A, the BOLD response to tutor song in the left and right sided midbrain are similar, but the responses to novel song differ between the hemispheres. In figure 3G, the tutor 2 bars are also similar in left and right MLd, but there's a difference in the tutor 1 and novel bars.

Thus, it would be incorrect to interpret the findings as a left-sided dominance for tutor 1, and a right-sided dominance for tutor 2.

That makes me think that you may want to change this figure and that the subheading in line 215 should be rephrased.

5) In figure 5B, the sequentially tutored birds are split up in good and poor learners. Now, the TUT2 > TUT1 contrast suddenly shows very different activation than in figure 3; how is that possible? In figure 3, the TUT2 > TUT1 contrast shows MLd activity, while the good vs poor learners show NCL or no activity.

6) The text in the discussion is a bit wordy and the line of thought is less clear than in the rest of the manuscript; I think you could make it more concise.

Minor comments:

- Line 61: Song similarity to tutor 1 was calculated relative to a novel song in the control group, and tutor 2 in the sequential group – but tutor 2 was specifically chosen to be very different from tutor 1. Can you use a novel song in the sequential group too, to ensure using a similar song control for the two groups?
- Figure 1C: In the 55 dph graph for sequentially tutored birds, why are there 7/16 individual points in the similarity to tutor 2 that are much higher than 0, but is the mean (red line) almost at 0?
- Line 99: "normally" is confusing, because the father was removed early
- Lines 102, 105: Is NCL generally thought of as part of the auditory lobule, or is it lateral to the auditory lobule?
- Figures 2B & 3B: The insets are too small to present the data.

- Figure 2B main image: There is a very large region of significant BOLD response over the left hemisphere. What do you think causes this? It seems weird to me not to mention and discuss this.
- Figures 2D, 3 (except 3A), 5: the color scale legends do not indicate any t-values other than 0, including the maximum t-value.
- I don't think the main text refers to figures 2C and 2E.
- Line 199: "emerged with age" > I would rephrase it to reflect that there was also more exposure to tutor 1 (not just growing older and getting more song practice).
- Why is the ROI in figure 4B so much bigger than the region of significant BOLD response in figure 3F?
- Figure 5A good learners: To what anatomical regions do the voxels with significant activation correspond? What do you think is the anatomical correlate of the biggest region of activity with higher t-values as indicated by yellow coloration (dorsal-lateral to the auditory lobule in the right hemisphere)?
- Figure 5A good learners: The auditory lobule is indicated – is it located in the right hemisphere or are there also significant voxels on the left? And this seems to be a small region, while the auditory lobule is big. Can you be more specific in the naming of the area? Or are other clusters of activation in this image also part of the auditory lobule?
- Line 262: awkward phrasing, "of the song from the song tutor"
- Lines 261-266: I cannot completely follow the sentence and it is quite long. Can you rephrase?
- Line 270: the explanation what NCL is seems out of place here; should have been earlier?
- Lines 296-299: not very clear. Do you mean that a strong response to novel song indicates plasticity? How and why?
- Line 301: What do you mean by "novel songs were processed differently"?
- Line 310: Could you test this by comparing the first blocks in your experiment with the last blocks? Also, this is a detail, but did you not play the song multiple times during each block? (5-6 bouts per 32 second stimulus)

Reviewer Comments

Reviewer #1 (Remarks to the Author):

The study of the team of Sharon Gobes found that acquisition of a second song (TUT2) changes lateralization of the auditory midbrain and that activity in the caudolateral Nidopallium (NCL), a region adjacent to the secondary auditory cortex, was related to the fidelity of second-song imitation. They demonstrated that during sensorimotor development, LMAN had stronger activation upon novel song exposure as compared to TUT1.

This is very nice work and very exciting data on zebra finches with clear relevance for human speech and second language learning.

Thank you for the kind words about our study; we appreciate that you find our data exciting and relevant.

These exciting findings however require some additional validation of the structures assigned to the data. To clarify: They claim involvement of LMAN, In figure 2, the activated region is huge using uncorrected data and one of these figures even shows Area X activation. To properly confirm this finding, one should show images of FWE corrected data in which the LMAN region should 'survive' and accordingly to priorly set criteria (eg if possible and as defined by literature: minimal cluster extend of 5 voxels). The data presentation would benefit from showing the outcome with cross hairs in 3 different orientations (or by even adding stereotactic coordinates eg according to Nixdorf or other).

We agree with the reviewer that the cluster at $p_{\text{uncorrected}} < 0.05$ observed in 55-day old juveniles is huge and also includes AreaX. As the reviewer suggested, we now used a more restrictive threshold of $p_{\text{uncorrected}} < 0.01$ and minimum cluster size of 5 voxels, followed by a ROI analysis at the level of IMAN and AreaX. ROI analysis at the level of IMAN and AreaX showed that BOLD activity specific for NOV songs was significant in IMAN ($p_{\text{FWE}} < 0.05$), while the activity in Area X approached significance ($p_{\text{FWE}} = 0.064$). As suggested by the reviewer, we have updated all the figures with crosshairs on the active clusters and have included sagittal sections, which should improve the readers' understanding of the exact anatomical location of activated clusters.

The same is true for fig 3 assigning NCM and field L which would also benefit from sagittal images with cross hairs. The same for midbrain, where MLd data are finally extracted ROI wise, where the corrected data should confirm the location of MLd. However LMAN and Field L, are still an easy target, as they are distinguishable with MRI contrast. The biggest hurdle however and the most interesting conclusion at the same time is the role of NCL which requires an accurate location of this structure (eg figure 5). The authors refer to the correct publication to verify this more carefully (von Eugen et al). Fig 9 in this publication shows the stereotactic coordinates and the section window (could be compared with yours) according to the Nixdorf atlas. The summarizing figure 12 in this publication illustrates how zebra finches do not have a single NCL structure but this structure is subdivided and spread over NCLd, NCLv and NCLm. Are you sure the activation is in an NCL subregion and not in NCC? Etc...Also here, displaying 'corrected' data in three different directions is mandatory and could lead to clear definition of NCL parts.

As mentioned above, we have updated all the figures with crosshairs on the active clusters and have included sagittal sections, indicating the exact location of the clusters. We are thankful for the reviewer's comment about matching MRI contrast with the article by Von Eugen et al. (2020) for distinguishing NCL. The NCL region that we found to be active in good learners is lateral to field L as shown in section number A1.35 of the Nixdorf-Bergweiler BE & Bischof HJ (2007) atlas, and is part of dorsal NCL (NCLd). Dorsal NCL in zebra finch is subdivided in rostral and caudal separated by a field of low TH + fibers (Von Eugen et al. 2020). The activated cluster in good learners is most likely dorsal NCL. However, the activated cluster in this study did not exactly match any of the subdivisions mentioned in Von Eugen et al (2020). The subregion of NCL active in our study lies lateral to field L and between field L and NCLdr described in Von Eugen et al. (2020). Therefore, we decided to refer to the active cluster as NCL. NCC active in female zebra finches (Ruijssevelt et al. 2017) is ventrolateral to field L, while the activated cluster in good learners lies in between NCC and field L (Fig 5B). We have now added line drawings of the atlas (A1.35 section; Nixdorf-Bergweiler BE & Bischof HJ, 2007) and the corresponding slice of zoomed in MR images where other contrasts are visible to identify the active NCL region in the current study in Fig 5C-I.

We have also updated our results using a more restrictive threshold of p-uncorrected < 0.01 and minimum cluster size of 5 voxels in Fig 5A-B.

Another remark on how the data are displayed is that when comparing activations upon different stimuli, one needs to display the same section(s) of the brain for each stimulus. In figure 3B for example, the bottom left picture has a different location and we cannot conclude on whether there were activations on this slice for the other stimuli. Drawing conclusions from the fMRI on specific regions is key: There is only one figure (fig 1) demonstrating location of regions and they are not mentioning the one referred to in the results (NCM, NCL, LMAN) and other locations than the one in figure 1 are used (fig 3B,H and 5B bottom left and fig 2 for LMAN). You should clearly overlay with atlas drawings in the results section what are the regions you assign. In conclusion, without stereotactic coordinates or contrast landmarks in the images, or 3D localization ...one can not assign a specific brain structure/region/nucleus to the activated regions. I think the authors can adjust this.

We thank the reviewer for this suggestion. We have now included crosshairs on the activated clusters in all figures, and also added sagittal sections to visualize the exact locations of the regions involved. The same sections are now shown for each stimulus. We have also added the corresponding line drawings of the parasagittal sections from the zebra finch histological atlas (Karten 2013) for comparison with the MRI atlas, as for the reader to better assess the locations of the activated clusters in the current study. These suggestions have been extremely helpful in improving the article and making it easier for the reader to interpret the figures.

Further remarks on results section:

Fig 1, legend: for consistency one should mention that red lines deal with NOV instead of unfamiliar

To clarify, we computed mean similarity scores between Tutee songs and a completely different set of unfamiliar/novel birds (not those that were used for NOV stimuli for fMRI). To avoid confusion, we have changed 'unfamiliar' to 'NOV' songs in the legend of fig 1.

93: in the legend of Fig 1: red horizontal lines indicate mean similarity with unfamiliar birds. This is confusing as in general and also in the figures NOV is used? Or did I miss something?

We agree that the interchangeable use of 'unfamiliar' and 'novel' is confusing and have changed 'unfamiliar' to 'NOV' everywhere in the text.

168:PFWE<0.1 : is not significant !

(There may be a typo in the reviewer's comment? Perhaps this was referring to PFWE = 0.051?). With the new analysis, this has now been changed to $p_{\text{uncorrected}} < 0.01$ with minimum cluster size of 5 voxels.

Figure 5: the color coding is a bit too dark for good learners as compared to the color used in the figure

Color coding was changed to the colors used in the graph in Fig 5 E-F.

Remarks on the material and methods section:

482: How is the animal positioned and fixed in a vertical bore system ? head up or down? a figure could help.

The birds were placed head up in the magnetic bore. To address this, we have added a picture in Supplementary Fig 4D showing the placement of the bird in a customized bird holder. The bird is placed in the probe after anesthetizing and the probe is placed with the bird's head up in the vertical bore.

482: 400 MHz instead of Hz

Made in-line edit '400 MHz'.

484: "Briefly" 1.5 – 2 % isoflurane. Authors do not specify whether this is initiation dose or also the maintenance level.

We used an induction rate of 2% isoflurane followed by maintenance at 1.5%. We have changed the wording in the text reflecting the different isoflurane flow rates to "Briefly, birds were anesthetized with isoflurane (induction: 2%; maintenance: 1.5%) in oxygen (99.5%) at a flow rate of 400cm³/min using a VIP Veterinary Vaporizer (Colonial Medical Supply Co., Franconia, NH)".

484: Two different flow rates are mentioned for isoflurane in oxygen. Specify the reason for the different flowrates and when each rate is used. Provide info on gas mixture % oxygen, % nitrogen or did one use pure oxygen? If so why? And are there publications doing this ?

The two different flow rates were an error taken over from a study from a senior thesis, but for this study we only used 400cm³/min (now changed in the text). Isoflurane mixed with 99.5% oxygen was used for anesthetizing the animal. We used an initial 2% Isoflurane for inducing the animal and later reduced the rate to 1.5% or as low as needed to maintain 60-100 breaths per minute for the bird as long as the bird was inside the NMR machine. We have also added the percentage of oxygen used in the text at line 484 "Briefly, birds were anesthetized with

isoflurane (induction: 2%; maintenance: 1.5%) in oxygen (99.5%) at a flow rate of 400cm³/min using a VIP Veterinary Vaporizer (Colonial Medical Supply Co., Franconia, NH)”

487: “the temperature inside the magnetic bore was maintained at 36°C ” did the authors determine the animal’s temperature? Birds have 39 to 40°C body temperature. Both temperature and breathing rate are important indicators of animal physiology and influence the BOLD response (affinity of Hb for O₂).

Respiration was monitored using a pneumatic pressure respiration sensor pad (Sims Graseby) placed underneath the subject’s chest and BioTrig Builder 1.01 software. We maintained the body temperature inside the magnetic bore as close to the bird's body temperature as possible in our system, using a water circulation unit which was maintained at 36 °C. Since the vertical bore of the magnet was only big enough to hold the bird, it was impossible to place a heating system around the bird. The bore could not be heated up above 36 °C, as this would damage the coils.

492: 3D rapid acquisition: Is it 3D? looks like multi-slice 2D acquisition since you mention 0.75mm interslice gap ? what is exactly interslice gap? Does it comprise slice thickness? Seems quite large in a high-resolution acquisition. The FOV is not mentioned

The reviewer is correct. It is multi-slice 2D acquisition and we have changed it in the text and have also added fov for both structural and functional scans. Both slice thickness and interslice gap distance is 0.75mm. Fov for functional scans as well as structural scans is 2.5 x 2.5 cm. We have also added this information in the method section at line 525 *FOV: 2.5 x 2.5 cm*.

497: Why did the authors choose such a high effective TE of 60ms ? What effect does it have on the images ? the paper does not show any fMRI data. Can the authors elaborate on which echo (of the 8) is used to calculate the effective TE?

We chose a high effective TE of 60 ms in combination with a RARE factor of 8 to get the best possible signal to noise ratio. In pilot studies we increased the RARE factor from 8, 12, to 16, and decreased the matrix size for acquisition while keeping the TE constant (unpublished data). We found the best results with TE effective = 60 ms and a rare factor of 8. Also, T₂ relaxation time for venous blood is less than 10 ms at 9.4 T, the relatively long TE associated with the SE pulse sequences allows for sufficient spin dephasing in large blood vessels, thereby removing their contribution to the BOLD signal and thus the BOLD signal can be largely attributed to extravascular BOLD signal from the small capillaries responding to an increased metabolic demand (Lee et al 1999; Poirier et al 2010). The 4th echo in the echo train was used to determine the effective TE.

We have added this text in the article at line 497 “Because T₂ relaxation time for venous blood is less than 10 ms at 9.4 T, the relatively long TE associated with the SE pulse sequences allows for sufficient spin dephasing in large blood vessels, thereby removing their contribution to the BOLD signal, which can be largely attributed to the extravascular BOLD signal from the small capillaries responding to an increased metabolic demand”.

499: Authors mention a scan duration of 120 minutes, is this total scan or fMRI scan duration. Under the paragraph “auditory stimulus presentation”, the authors discuss two separate acquisition schemes for 55dph and 90dph, which consist of two block designs of different sizes.

120 minutes is the fMRI scan duration for 90-day old birds where 450 scans were taken and 80 minutes for 55-day old birds where 300 scans were taken. We have modified the text in the article to reflect this information. (“fMRI scan duration: 80 min (55 dph) or 120 min (90 dph)”)

511: The authors give more detailed information on their block design stimulation scheme. However, when calculating the scan time, this does not match with what has been mentioned in line

499 (120 minutes). 55dph; 300 repetitions = 80min, 90 dph; 450 repetitions = 120 min

We are thankful for the reviewer catching the discrepancy. We have corrected it in the text at line 499 and now reads “fMRI scan duration: 80 min (55 dph) or 120 min (90 dph)”

512: What is the reason for the 1:2 ratio for ON:OFF blocks? Can the authors motivate this decision with literature or have additional experiments been performed to assess its benefits? Reducing the OFF-block duration would result in a dramatic decrease in acquisition time which is cost beneficial and reduces the anesthesia time for the animal.

In addition to the regular 1:1 ON/OFF blocks (32 sec each), we added 2 dummy scans after the OFF blocks but ultimately did not use these for the analysis. We agree we could have reduced acquisition time had we not added extra blocks which is something we will be taking care of in future studies. We have made changes to the text at line 512 “Auditory stimuli were presented in a fixed order in an ON/OFF block design consisting of 32 seconds of auditory stimulation (ON) followed by 32 seconds of silence (OFF). Two additional dummy scans (32 seconds) were added after every OFF block but were not used for analysis. Each fMRI scan is 16 seconds long and two fMRI scans make up one block. An fMRI session for a 55-day old bird consisted of 50 ON blocks (25 per stimulus, TUT1 | NOV) and 50 OFF blocks, and an fMRI session at 90 dph consisted of 75 ON and 75 OFF blocks playing three stimuli (TUT1 | TUT2 | NOV) in the same order”

514: Playing 3 stimuli in the same order... Normally this is randomized to reduce expectations/prediction and habituation to the stimulus. Why did one not choose to do this?

The reviewer raises an excellent point. We agree that the randomized stimulus order would have been better in reducing expectations and habituation. However, the two additional dummy scans after the silence period add extra time for the animal to recover from a heightened attentional state due to the last stimulus played. Because of habituation, we may have missed some potential results, but it is unlikely that we gained effects that would not have been there with a randomized order. For future experiments, we will include a randomized stimulus order.

516; resulting in 50 images per stimulus type per subject: for which age group? Explain this number?

To clarify, all animals in each age group (55 or 90dph) received 2 fMR scans during each ON block. Each stimulus was presented 25 times for a total of 50 images per stimulus.

Line 494: “Two T2–weighted RARE images were acquired during each block, resulting in 50 images per stimulus type, per subject at 55 dph and 50 images per stimulus type, per subject at 90 dph.”

526: How was head motion assessed ? From realignment read-out or visually assessed before realignment? Was only translation or also rotation considered?

Head motion was assessed from the realignment readout which gave us values for head motion for translation and rotation and both were considered to assess head motion in all the animals.

Line 561: “The following inclusion criteria were applied to the fMRI time series: 1) limited head motion (< 0.5 mm translation or rotation in any of the 3 directions) as measured from the realignment read out given by the SPM realignment step.”

540: Add matlab version

MATLAB version R2015b was used for all the preprocessing steps mentioned in the article and it has been added to the text.

536 – 547: Different transformation steps are mentioned. Were these transformations performed separately with different reslicing steps? If so, why weren't the separate transformations combined to prevent the (potential) additive bias of consecutive resampling steps?

We realigned functional images and then co-registered it to the 2D-multi-slice RARE anatomical scan of high resolution of each subject using FLIRT, which was linear and maintained proportions. Final transformation to functional images was applied in the last step using a downsampled atlas (dim: 64x64) as reference image. For clarity we have rephrased the method sections and it now reads:

“fMRI time-series scans from each subject were realigned to correct for intra-individual head motion using a six-parameter rigid body spatial transformation (without reslicing) in SPM12. fMRI scans were masked to exclude non-brain tissue, and a Gaussian Kernel of 0.5 mm full width at half maximum (FWHM) was applied to achieve in-plane smoothing using custom written scripts in MATLAB (R2015b). The realigned images were then co-registered to their own 2-D multi-slice RARE anatomical scan using affine registration with the FLIRT tool in the data processing software “FMRIB Software Library” (FSL, <https://fsl.fmrib.ox.ac.uk/fsl/>). In parallel, the 23-D multi-slice RARE anatomical scan of each subject was spatially normalized with the high resolution zebra finch MRI atlas ²⁴ with ANTs using mutual information as similarity metric (<http://stnava.github.io/ANTs/>). The transformation matrix generated in the previous step was then applied to the realigned and co-registered functional data using ANTs apply transform (<http://stnava.github.io/ANTs/>), resulting in functional data co-registered to the atlas. The down-sampled atlas was used as a reference point in the previous step for warping functional images to the atlas space which kept the functional image resolution at 64X64X16.”

547: “Precisely co-registered to the atlas” The authors do not specify the final resolution of the functional MRI data. Using the terminology “precisely co-registered” could be interpreted as an upsampling step to the high-resolution zebra finch MRI atlas.

We agree with the reviewer, using ‘precisely co-registered to the atlas’ gives the readers a wrong impression of the functional data to be upsampled. That was not the intention and have removed that in the text to instead read ‘co-registered to the atlas’.

558: all images 'where' acquired > should be 'were'

'where' has been changed to 'were'. Thanks for catching that.

574: Whole brain voxel-based analysis is performed without family-wise error control. While the authors claim that this is too conservative for small animal fMRI, the results shown in the paper illustrate clusters that survive FWE correction.

Why is the very liberal threshold of $p_{\text{uncorrected}} < 0.05$ used instead of a more restrictive $p_{\text{uncorrected}} < 0.001$ or smaller which is better to control for false positive results?

As the reviewer suggested, we have updated figures showing corrected clusters. Our significance threshold was $FWE < 0.05$. FWE corrections were performed within the clusters that were found to be active in the main effect of the analysis, and reported with t_{max} and p_{FWE} if significant and if not, we have reported t_{max} and the uncorrected p -values. In Fig 2 we performed an ROI confirmatory analysis with FWE correction.

Uncorrected data can be used exploratory and corrected data listed in table (eg peak voxel intensity and number of voxels). When claiming involvement of (new) regions one should test whether they survive correction

We agree that there are more false positives with a $p_{\text{uncorrected}} < 0.05$ threshold. Therefore, we have only used it as an exploratory threshold for one-sample t -tests in Fig 2. Although the NCL region activated in good learners does not survive the FWE correction at the whole brain level, the combination of an adjusted threshold of $p_{\text{uncorrected}} < 0.01$, and the significance of the correlations with similarity scores points at a biologically relevant result. With these stricter criteria, we have 'lost' the NCL cluster in the right hemisphere. We don't think that only the left NCL is involved in 2nd tutor song learning, but this results from the sensitivity of the method and that the BOLD effects are strong enough in the left hemisphere but not in the right.

627: could not be delineated in the MRI atlas due to their size. It is more because they did not have a differential contrast in the images. Ov is indeed too small. Overall one could co-register with an atlas though....

We thank the reviewer for pointing out this discrepancy in our writing that delineation of VTA and SNc is difficult not just because of their size but because those regions lack differential contrast in the atlas. We have added this to the text at line 627 "because of the lack of a differential contrast in MR images"

Remarks on discussion section:

312: Another factor that could confound our results is the auditory isolation of birds in between tutoring sessions. Then an explanation follows which should overrule the potential effect of isolation. This explanation is not clear to me. Also the isolation between 32dph and 55dph, is there any literature information on whether this prolongs the sensory period? The sensitive/critical period? keeping some brain structures/nuclei sensitive for neuroplasticity for a prolonged period? and influencing the impact of TUT2? Why did you actually choose this specific experimental setup with acoustic isolation periods in between? All this deserves a place in the discussion.

The reviewer has raised an important point that the isolation period is quite long and could confound our results. We based our experimental paradigm on the work of Yazaki and Mooney (2004) who followed a similar protocol for sequential tutoring of zebra finches. Social isolation extends the sensory acquisition period beyond 60 dph (Livingston 2000), which is critical for this specific experiment with two tutors (it may allow for the plasticity necessary to copy syllables from a second tutor). Young male siblings influence each other's songs in the absence of adult males (Derégnaucourt and Gahr 2013) and females also provide feedback to male juveniles during the sensory period (Carouso-Peck and Goldstein 2019). We chose to keep the birds isolated in between the two tutoring periods to ensure that they got to practice their songs in the absence of any social feedback.

We have added this in discussion at line 312 “On the contrary, the social isolation between 32-55dph may have extended the sensory acquisition period (Livingston 2000, Yazaki and Mooney 2004)”.

Finally I would change the title in the direction of: Tracing development of song memory with fMRI in zebra finches after a second tutoring experience. Maybe mention also the sensorimotor stage? I leave it up to the authors.

We have followed the reviewer's advice and have changed the order of 'with fMRI', but because of the 15-word limit for the title have left the wording of the title as-is.

Reviewer #2 (Remarks to the Author):

In this study, the authors show that fMRI-based neural activity is associated with tutor song memory when juvenile zebra finches were sequentially tutored by two adults during the sensitive period of song learning. Behaviorally, juvenile finches were able to copy the second tutor song after the song memory acquired earlier from the first tutor. Some of the neuroimaging results are really interesting, the tutor song memory is identified in NCM and NCL, and the changes of lateralization in midbrain after the second tutor presentation. This study provides important findings about the neural basis of song memory acquisition during the sensitive period of sensorimotor learning. This manuscript is clearly written, and the statistics are valid. However, a few major issues need to be clarified.

Thank you for the kind words; we appreciate that you find our study really interesting and that it provides important findings.

1. For a fMRI study like this, a well characterized and quantitative behavioral measure is crucial. The authors should provide a more detailed description and examples of (1) selection and quantification of song similarity between tutors; (2) quantitative measure of percent similarity between tutor and tutee.

Quantitative measure of percent similarity between the first and second tutors (and a novel conspecific song). How many adults (sample size) were used as the first tutor and as the second tutor? If the second tutors were chosen based on low similarity between the first and second tutors, what is the “low” similarity quantitatively? Do the first and second tutors have similar duration of motif (or the same number of syllables per motif)? What specific function of similarity match was used in SAP for this quantification (symmetric or asymmetric scoring)? Can authors provide sonogram examples of the first and second tutor songs (and novel songs) in Figure 1 or a supplementary figure?

We used fourteen adult males from the colony as first tutors and another set of 14 birds that were used as second tutors. We used asymmetric measurements for finding similarity between songs. There was no significant difference between motif duration between the two tutors (TUT1: 1.03 ± 0.23 (mean \pm SD) seconds; TUT2: 0.85 ± 0.18 seconds; two sample-tests, $p > 0.05$). The number of syllables in TUT1 was 6 ± 0.7 (mean \pm SD), and the number of syllables in TUT2 was 5 ± 1.2 . We looked at cumulative % similarity between the two songs and less than a 50% match between two songs was considered low similarity. This has all been added to the manuscript. Based on the suggestion, we have also added sonograms for two representative tutees with their TUT1 and TUT2 in Supplementary Fig 3A-B.

Quantitative measure of percent similarity between a tutee and its first or second tutor. I agree that it is a good idea to use two blind judges to observe and identify the song similarity score between a juvenile and its first or second tutor. To better visualize the similarity measure, it is a good idea to provide sonogram examples of a good similarity match to the first tutor vs. the second tutor (as well as novel songs). Was sequential order of syllables (or syntax structure) used for this quantification?

As the reviewer suggested, we have added sonograms of tutees and their TUT1 and TUT2 songs when they were 55 and 90 day old in Supplementary Fig 3A-B. We did not take into account the sequential order of the syllables for similarity. We scored syllables based on how well they matched with each other.

2. Social isolation. In this study, juvenile finches were socially isolated from 32-55 days of age (more than 3 weeks of social isolation?) then again from 66-150 dph. Why placed the juveniles in isolation for so long? I guess the 3-week social isolation (and then provided a second tutor at 55 dph) probably could motivate isolated juveniles to interact with the second tutor and the juveniles are more likely to learn from the second tutor? Social isolation is also known to delay song development in zebra finches (Morrison, 1993), therefore this isolation may extend neural plasticity for song acquisition from the second tutor? What happened if a juvenile was exposed to the first tutor then to the second tutor without social isolation? It would be great to have a non-isolated group as a control.

We agree with the reviewer, social isolation in between tutoring periods could have an effect on the neural plasticity in the brain and eventually influence learning from the second tutor. We decided on a social isolation period before providing the second tutor, in order for the birds to learn and practice the song of their first tutor and not be influenced by their male siblings (Derégnaucourt and Gahr 2013) or feedback from females (Carouso-Peck and Goldstein 2019). However, we think a control group of birds re-exposed to their TUT1 after a similar isolation period was more appropriate in answering if the experience with a second tutor during the sensorimotor learning period affects song learning and changes neural activity.

A non-isolated group would have been a good addition to the experiment, but keeping the logistics of doing longitudinal MRI in multiple birds manageable, we chose to include a control group which was treated the same as the sequentially tutored birds except for the kind of tutoring they received.

As the authors noted, zebra finches are social and colonial birds. I understand social isolation is a relatively common experimental procedure for neuroethological study of birdsong learning, but several weeks of isolation seems quite dramatic for a social animal, especially the isolation (32-55 dph) was performed at the peak of the sensitive period for song learning in zebra finches. Could such “unnatural” social isolation delay or affect the developing auditory system,

hemispheric differences in tutor-selective midbrain activity, or LMAN selectivity response (as previous electrophysiology or neuroimaging studies showed different results) during the sensitive period of song learning? For the future study, visual isolation or alternative methods (housing with a juvenile female bird, for example) might be more appropriate.

We agree with the reviewer that social isolation for several weeks can be quite dramatic for social birds. However, we chose to keep the birds isolated from other juvenile male siblings because they could influence each other's songs in the absence of an adult tutor (Derégnaucourt and Gahr 2013); songs of juvenile male finches are also influenced by females in the clutch (Carouso-Peck and Goldstein 2019). We based our experimental paradigm for sequential tutoring on a previous study in zebra finches that were sequentially tutored during the sensorimotor learning phase by adult Bengalese finches (Yazaki and Mooney 2004). In this study, electrophysiology showed TUT specific activity in IMAN after experience with the first tutor, and the same adults showed TUT2 specific activity in IMAN after tutoring with the second tutor. Social isolation in between tutoring periods does extend the sensory learning period (Livingston 2000). If the activity in IMAN or MLD is due to acute isolation, we would have also observed that at 90 dph because birds were isolated for the same amount of time after exposure to TUT1(32-55dph) and TUT2 (60-90 dph). That was not the case, as the BOLD response to NOV songs in IMAN was specific for juveniles and was not observed in adults.

Minor issues:

1. Figure 1c. The behavioral results are quite interesting, I wish the authors can elaborate more of the behavioral results in this figure or in a supplementary figure. They can help us to better understand the neuroimaging results. What does each dot represent? It is interesting that some of the juveniles had a high similarity match (almost 100%) to TUT1 at 55 dph. Did this bird match well to TUT1 at 90 dph, or it modified its song syllables and syllable order (syntax) and matched TUT2? Was a good learner for TUT1 also a good learner for TUT2? Or does good learning from the first tutor impose constraints to learn from the second tutor? It would be nice to visualize the song imitation by showing sonogram examples of crystallized songs with high and low similarity matches.

Most of the birds (14 out of 16) learned new syllables from the second tutor and added them to the song that they were singing at 55 days old. Each dot represents an individual bird. The particular bird that had 100 % similarity at 55 dph, added some syllables from the second TUT but still had higher similarity with TUT1 at the end of the learning period. With the patterns so variable, and the number of birds we looked at, we could not discern a pattern of first- vs. second-song learning behavior (e.g. good first remains good first, or poor first becomes good second, or good first becomes good second). We did not look at the syntax as it was hard to determine this in 'mixed' songs which are made up of first and second tutor syllables. We have not explored the behavioral reasons behind why some birds learn well from the second tutor, or taken into account features of TUT1 that could be a driving factor in enhanced learning. However, these are exciting questions that we would like to explore in future studies.

As the reviewer suggested, we have added examples of tutee songs with their corresponding TUT1 and TUT2 songs at 55 and 90 dph in Supplementary Figure 3A-B.

2. Line 436, 32 male finches are from how many clutches?

We raised 25 clutches which gave us 32 males for the study (added to the manuscript).

3. Line 457, song motifs were selected between 2pm-11pm. 11pm seems to be really late for a diurnal species. What was the onset of the (16:8) light cycle? What was the reason to have a light cycle like this?

The light cycle was set at 10 am (on) - 2 am (off), 16:8 L:D, with the last light hour characterized by gradually decreasing light intensity resembling dusk conditions. Zebra finches are commonly found in central Australia in an environment characterized by long days in summer (15 hrs sunrise-sunset). We maintained our breeding pairs in these conditions because they mimic naturalistic conditions and optimize breeding outcomes, and hence the offspring from these breeding pairs were also kept under the same conditions as to not change the light cycle mid-experiment. The late onset (10 am) was chosen to work well with experimenter schedules, but as their only light source is artificial, this does not affect the birds.

4. Line 504, Conspecific songs or Novel conspecific songs?

In line 504, the third stimulus played in MRI is novel conspecific songs.

5. Line 507. 5-6 bouts of song recordings were from the repetition of the same song bout or 5-6 different song bouts (control of pseudoreplication)?

We thank the reviewer for pointing this out. We controlled for pseudoreplication and used 5-6 different song bouts. We have also changed it in the text so there is no ambiguity “5-6 different bouts of song with 1-2 seconds...”

6. Discussion: Parallel with human second language. The authors discussed the parallel between birdsong learning from the second tutor and human second language learning. Although the comparison of hemispheric dominance between birdsong and language learning is interesting, there are limitations for this comparison. Zebra finches crystallize only one song. They are genetically “monolingual” and are not able to acquire, process, and produce two different song types sequentially and simultaneously. Male finches have the learning plasticity to modify their previously memorized song from the first tutor to match the later exposed second tutor during the sensitive period and end up crystallizing one song type for the lifetime. Also, the animals in this study were under “unnatural” isolation manipulation. The underlying neural mechanism could be very different from sensorimotor processing of second language learning.

We agree with the reviewer’s point that the experimental birds were raised in ‘unnatural’ isolation conditions. We raised both the sequentially tutored birds and the control birds with the same number of isolation periods in between tutoring sessions to be able to compare the neural activity associated with acquiring attributes from two different tutors. We did not intend to imply that the neural mechanisms that govern this (unnatural) plasticity in the zebra finch brain, allowing it to learn new features of a second song, are comparable to the mechanisms of second language learning. We merely want to point out that even though these species are far apart from each other, they share certain similarities in the neural activation pattern related to vocal plasticity. The isolation period was important to control for other auditory and social input, which would not be possible in human studies where it is hard to determine which factors drive plasticity related to second language learning.

We have now discussed this caveat in the discussion (line 476)

“Even though zebra finches can acquire new syllables from a second tutor during the sensorimotor learning period, they, unlike humans, cannot flexibly switch between the two songs

in adulthood. While the neural mechanisms of using and producing a second language or second song could be quite different, our results indicate that the strikingly similar neural patterns we report for birds could be related to the additional neural plasticity necessary for auditory-vocal learning late in development.”

Reviewer #3 (Remarks to the Author):

This is an interesting paper addressing the neural substrate for second tutor song memory, as a parallel for second language learning. I have some comments on and questions about the presentation and interpretation of the findings, but I think the authors should be able to address these comments without too much additional work.

Thank you for the kind words, we are happy that you find our paper interesting.

General comments:

1) Can you please add a figure with song examples of both groups on 55 and 90 dph? I would like to get a better sense of the level of song imitation.

We have added 2 examples of tutee songs at 55 and 90 days with their TUT1 and TUT2 songs for comparisons in Supplementary Fig 4A-B. In this figure, we have indicated syllables copied from TUT1 (red bars) and TUT2 (blue bars).

2) The regions with significant BOLD response are often much bigger than the anatomical regions that you subscribe the activation to (e.g., LMAN or MLd). Why do you think this is? Is the response not specific to the song nuclei? Is there a lot of anatomical variation between the birds? Something else?

Initially, we used an exploratory threshold (p -uncorrected < 0.05) to display the regions of activation and with this non-restrictive threshold the clusters are bigger than some of the anatomical areas. We have now used a more restrictive threshold of p -uncorrected < 0.01 with a minimum cluster size of 5 voxels. We have also included ROI analysis using anatomical ROIs for LMAN and MLd and showed that those regions are active after FWE correction. Alternatively, the regions surrounding MLd in Fig 3 A-B may correspond to smaller regions like the substantia nigra (SNc), the ventral tegmental region (VTA), and the intercollicular nucleus (ICo) which cannot be delineated in MR images because they lack contrast. This may be also the case for LMAN 'shell'.

3) Do you want to add an explanation about what neural processes are thought to be reflected by positive vs. negative BOLD responses? Some of your differences are caused by more negative BOLD responses (rather than by higher positive BOLD responses). For example, I think the poor learners in figure 5 have more negative responses in D than C, haven't they? Or 3G: right TUT2 is significant compared to slightly negative NOV and TUT1 responses; while in the left MLd there's a similar pattern but NOV and TUT1 are not negative and TUT2 is not significantly different.

The reviewer raises an excellent point. We observed differences in songs in MLd which may be a result of negative BOLD responses to NOV songs in Fig 3G and 3E. We have added an explanation for the negative BOLD responses at line 325 of the Discussion:

“Hemispheric differences observed in MLD in Fig 3G and 3E could also be attributed to differences between songs due to negative responses while listening to NOV songs. Positive BOLD responses reflect increased neural activity, but negative BOLD responses could either be because of decreased neural activity or “vascular steal” . The BOLD responses to NOV songs in MLD are characterized by large variability between birds and may not be robust enough to speculate about the underlying neural processes that could cause a negative BOLD response. However, it is interesting to note that both TUT1 and TUT2 show positive BOLD responses in the same region where NOV songs show no BOLD response or a negative BOLD response. This may represent neuroadaptive changes associated with learning, through which the TUT songs become more salient because of familiarity, while at the same time the sensitivity towards NOV songs is reduced.

4) It seems tricky to present the data as you do in figure 4, because now it may be tempting to interpret the midbrain lateralization result as a tutor-specific effect. However, it does not seem to be caused by a different response to tutor song in the two hemispheres, but by a different response to novel: In figure 3A, the BOLD response to tutor song in the left and right sided midbrain are similar, but the responses to novel song differ between the hemispheres. In figure 3G, the tutor 2 bars are also similar in left and right MLD, but there’s a difference in the tutor 1 and novel bars.

Thus, it would be incorrect to interpret the findings as a left-sided dominance for tutor 1, and a right-sided dominance for tutor 2.

That makes me think that you may want to change this figure and that the subheading in line 215 should be rephrased.

We are thankful to the reviewer for pointing out the differences in TUT and NOV responses in the midbrain when comparing control and sequentially tutored birds, which made us look at our results in a new light. We agree with the reviewer that it would be incorrect to interpret the findings in Fig 4 as left-dominant response in control birds and right-dominant for sequentially tutored birds. It would indeed be more appropriate to say that even though the BOLD responses to TUT songs in Fig 3A and Fig 3C are similar in the left and the right midbrain, there is a change in response to NOV songs between the hemispheres. With different tutoring experiences, BOLD activity for novel song in the midbrain decreases as the birds mature. As a result, the differential response for TUT1 over NOV songs in control birds is localized in the left midbrain and TUT2 over TUT1 songs in sequentially tutored birds is localized in the right midbrain.

The text at line 218 now reads as “In control birds, significant differences in BOLD responses to TUT1 and NOV songs (TUT1 > NOV) in the left hemisphere emerged after re-exposure to TUT1 with age as the birds reached adulthood (Fig 4A: 90 dph: left vs right, $t_{(12)} = 3.9$, $p = 0.019$; 55 dph: left vs right, $t_{(12)} = 0.877$, $p = 0.39$). In contrast, in the sequentially tutored birds, TUT2 selective responses also emerged with age but were localized in the right hemisphere (Fig 4B: 90 dph, left vs right, $t_{(14)} = -3.04$, $p = 0.008$; 55 dph, $t_{(14)} = 0.673$, $p = 0.51$). These differences in hemispheric selectivity for tutor song are due to a more negative BOLD response to other songs (NOV in the left MLD in control birds, and TUT1 and NOV in the right MLD of sequentially tutored birds) that emerged at the end of development. This ROI analysis thus confirms that tutor song selectivity is localized in the midbrain region at the end of the sensorimotor learning period, but the hemispheric differences depend on the learning experience.”

Also changed the subheading at line 188 “**Hemispheric differences in auditory midbrain activity depend on learning experience**”

5) In figure 5B, the sequentially tutored birds are split up into good and poor learners. Now, the TUT2 > TUT1 contrast suddenly shows very different activation than in figure 3; how is that possible? In figure 3, the TUT2 > TUT1 contrast shows MLD activity, while the good vs poor learners show NCL or no activity.

In Fig 3B, the voxels are shown that have a significantly higher BOLD response for TUT2 over TUT1 in all birds. When we split the same group of birds into two groups (good and poor learners), voxels with a higher BOLD response for TUT2 > TUT1 are activated significantly in NCL. These activated voxels in NCL would not show up as significantly activated in the analysis with all the birds, because the results are averaged over all the birds and half of the birds (poor learners) do not have activity in NCL.

6) The text in the discussion is a bit wordy and the line of thought is less clear than in the rest of the manuscript; I think you could make it more concise.

We have removed several parts of the discussion, and rewritten the text based on suggestions from all reviewers, including, for example, a discussion of the negative BOLD response. This has made the discussion more concise and hopefully clearer.

Minor comments:

- Line 61: Song similarity to tutor 1 was calculated relative to a novel song in the control group, and tutor 2 in the sequential group – but tutor 2 was specifically chosen to be very different from tutor 1. Can you use a novel song in the sequential group too, to ensure using a similar song control for the two groups?

To clarify, we compared both the control group and the sequentially tutored birds with a completely different set of novel conspecific songs at 55 and 90 days (means represented by the red lines in Fig 1 C).

- Figure 1C: In the 55 dph graph for sequentially tutored birds, why are there 7/16 individual points in the similarity to tutor 2 that are much higher than 0, but is the mean (red line) almost at 0?

The red lines in this figure are indicative of mean song similarity with an entirely different set of unfamiliar birds (not used as first or second tutors), and do not represent the mean of the similarity between tutees and their tutors. To avoid any confusion, we have changed ‘unfamiliar’ to ‘novel songs’ throughout the text and figure legends. In the figure legend, we have now changed this to “Red horizontal lines indicate mean similarity with an entirely different set of novel birds (different from those used as second tutors, which are also novel at 55 dph).”

- Line 99: “normally” is confusing, because the father was removed early

Changed it to “At that point, all animals (N = 27) were reared with their family (other nestlings and mother) and with their biological father as the first tutor (TUT1) up till 32 dph”.

- Lines 102, 105: Is NCL generally thought of as part of the auditory lobule, or is it lateral to the auditory lobule?

NCL is not considered part of the auditory lobule. It is homologous to the prefrontal cortex in mammals and has recently been shown to be active in song perception in zebra finches (Bottjer 2010, Ruijssevelt 2017). This sentence was confusing and we have now tidied it up.

- Figures 2B & 3B: The insets are too small to present the data.

We have removed the insets and changed the images.

- Figure 2B main image: There is a very large region of significant BOLD response over the left hemisphere. What do you think causes this? It seems weird to me not to mention and discuss this.

The large region in the Fig 2B (NOV > Rest) is quite lateral, may enclose the nidopallium region, but has not been defined in the atlas. Furthermore, we chose to show exploratory one-sample t-tests in Fig 2B to see if the songs activated auditory regions against rest periods (to fulfill the criteria for successfully detecting any activity in auditory regions as a result of auditory stimulation), but we only discussed clusters in the results section that were selective for one song over the other (TUT > NOV or NOV > TUT) using paired t-tests with a more restrictive threshold.

- Figures 2D, 3 (except 3A), 5: the color scale legends do not indicate any t-values other than 0, including the maximum t-value.

As the reviewer suggested, we have added values to all the color scale bars representing the range of t-values between 0 and maximum t-value

- I don't think the main text refers to figures 2C and 2E.

The reviewer is correct. We did not refer to the bar graphs in Fig 2C and 2E in the original document. We have now updated the figures and the text and made sure all panels are discussed in the text.

- Line 199: "emerged with age" > I would rephrase it to reflect that there was also more exposure to tutor 1 (not just growing older and getting more song practice).

We have made changes at line 210 "emerged after re-exposure to TUT1 as the birds reached adulthood", that reflects the hemispheric differences in control birds could be a result of re-exposure to TUT1 as well as vocal practice as they reach adulthood.

- Why is the ROI in figure 4B so much bigger than the region of significant BOLD response in figure 3F?

We calculated lateralization of responses from the clusters in Fig 3A and 3B, which are the results of the paired t-tests. The cluster in 3F represents an ROI restricted to MLd, while for Figure 4B we used a functional ROI. We have also added this information in the figure legend for Fig 4

- Figure 5A good learners: To what anatomical regions do the voxels with significant activation correspond? What do you think is the anatomical correlation of the biggest region of activity with higher t-values as indicated by yellow coloration (dorsal-lateral to the auditory lobule in the right hemisphere)?

We think the cluster dorso-lateral to the auditory lobule is part of the nidopallium and includes but is not limited to NCLm (as described in Von Eugén et al. 2020). This particular cluster does not survive FWE correction but approaches significance ($t_{\max} = 7.7$, $p_{\text{FWE}} = 0.08$, or $p_{\text{uncorrected}} < 0.0001$). We have added this information at line 214 which now reads “good learners showed increased activation for TUT2-song relative to poor learners in the nidopallium ($t_{\max} = 5.9$, $p_{\text{FWE}} = 0.08$ or $p_{\text{uncorrected}} < 0.0001$), which included but was not limited to NCLm (Fig 5C; Von Eugén 2020),”

- Figure 5A good learners: The auditory lobule is indicated – is it located in the right hemisphere or are there also significant voxels on the left? And this seems to be a small region, while the auditory lobule is big. Can you be more specific in the naming of the area? Or are other clusters of activation in this image also part of the auditory lobule?

The reviewer is correct that the auditory lobule is bigger than the significant voxels. We used ‘auditory lobule’ as a collective term for the NCM, CMM and field L. We used a restrictive threshold of $p < 0.01$ with a minimum cluster size of 5 voxels and found a small cluster in the nidopallium of the right hemisphere of good learners. We have added this information at line 214 in the text.

- Line 262: awkward phrasing, “of the song from the song tutor”

We have changed it to ‘of the song from the tutor’.

- Lines 261-266: I cannot completely follow the sentence and it is quite long. Can you rephrase?

We have now broken up this the sentence in the following way:

“In zebra finches, the NCM, a secondary auditory region, may even encode perceptual memories of the song from the tutor to which the bird was exposed early in life. In this region, expression of immediate early genes, rates of neurogenesis, and habituation of neural responses have all been shown to be related to the strength of song learning^{20, 30, 31, 32, 33, 34, 35}. Pharmacological inhibition or lesioning of the NCM impairs song learning and reduces the behavioral preference for the tutor’s song^{36, 37, 38}”

- Line 270: the explanation what NCL is seems out of place here; should have been earlier?

We have now added this to the last line of the introduction.

- Lines 296-299: not very clear. Do you mean that a strong response to novel song indicates plasticity? How and why?

In streamlining the Discussion, this sentence has now been removed.

- Line 301: What do you mean by “novel songs were processed differently”?

For clarification, we have changed this to “the higher BOLD response to novel conspecific songs indicates that there are brain regions which differentiate between learned and novel songs,”

- Line 310: Could you test this by comparing the first blocks in your experiment with the last blocks? Also, this is a detail, but did you not play the song multiple times during each block? (5-6 bouts per 32 second stimulus)

We thank the reviewer for pointing out the error. We have changed this to “which is measured across 25 stimulus blocks for a total of approximately 150 repetitions of the tutor’s song”.

Because of the small signal to noise ratio in BOLD responses, our block design with 25 blocks for each stimulus provides a good signal that can be statistically analyzed. If we were to compare only the first few blocks with the last blocks, we would not have a good enough signal to be able to find any effects (in this case, of habituation).

Details of changes made to the figures

Fig 1: As reviewer # 2 mentioned, some of the dots in panel C were not visible. The new graph shows all 16 data points.

Fig 2: As reviewer #1 suggested we added sagittal sections in panel A-F. We redid the analysis with a more restrictive threshold of $p < 0.01$ and minimum cluster size of 5 voxels. For paired t-tests in C-D, we have now presented voxels with $t > 2.47$, $p_{\text{uncorrected}} < 0.01$. We added a region of interest analysis for IMAN, using a ROI created from the zebra finch MRI atlas in panel F. The bar graph in panel G represents the average BOLD response (β weights) for each stimulus relative to rest periods within IMAN. Data shown as mean \pm standard error of the mean (SEM) across voxels in the cluster ($*p_{\text{FWE}} < 0.05$, $N = 28$). Line drawings of parasagittal sections from the zebra finch histological atlas (Karten 2013) were added in panel E, and detailed locations of activated clusters were added in H-I.

Fig 3: We rearranged the figure with panel A showing post-hoc paired t-tests in sequentially tutored birds. Statistical maps of voxels with $t > 2.46$ ($p_{\text{uncorrected}} < 0.01$) are displayed. We added line drawings of parasagittal sections from the zebra finch histological atlas (Karten 2013) in B and H, and detailed locations of activated clusters were added in C and I. We moved the statistical map of voxels for the comparison of sequentially tutored $>$ control to supplementary figure 3. We replaced the statistical maps in the ROI analysis obtained with up-sampled images (128x128x128)

Fig 4: We analyzed the data using a restrictive threshold of $p_{\text{uncorrected}} < 0.01$ and a minimum cluster size of 5 voxels, and thus we recalculated % signal change in ROIs in control (Fig 4F) and sequentially tutored birds (Fig 4A), which were based on clusters activated in control birds (Fig 3F) and sequentially tutored birds (Fig 3A) and their mirrored counterparts in the other hemisphere.

Fig 5: Good learners show selective response in NCL after learning the second tutor’s song, as compared to poor learners. A-B) We replaced statistical maps with the ones obtained at $p_{\text{uncorrected}} < 0.01$ and minimum cluster size of 5 voxels. C) We added a line drawing of a coronal section (A1.35) from the zebra finch stereotaxic atlas. D) detailed view of the clusters activated in A-B are shown with white dotted lines. E) β weights extracted from the updated clusters in (B) and the mirrored counterparts in other hemispheres and bar graphs and correlation in F-G were updated as well.

Supplementary Fig 1: No change was made in this figure

Supplementary Fig 2: Previous statistical maps showing auditory lobules in good learners at 55 dph was reduced to a small cluster in CMM and a second cluster in field L which is now included in panel A, as well as the β -weights extracted from the updated clusters in (B). We

removed the correlation graphs as the previous analysis is not valid anymore with the new clusters.

Supplementary Fig 3: We added a new supplementary figure showing some clusters which did not pass the threshold but are biologically relevant. A&B) Statistical map of voxels ($p_{\text{uncorrected}} < 0.05$) showing a cluster in NCM in the TUT1 > NOV contrast, and a cluster in NCM/NCL in the TUT2 > NOV contrast in sequentially tutored birds. C&E) Comparison between sequentially tutored and control birds ($p_{\text{uncorrected}} < 0.01$ with no minimum voxel size) and updated β weights extracted from the clusters shown with a white arrow in C&E. D&F) line drawing of a parasagittal section from the zebra finch histological atlas (Karten 2013) in D, and detailed location of the activated cluster in F.

Supplementary Fig 4: On reviewer's # 2 suggestion, we added a new figure. A-B) Representative sonograms of tutees in the control group (A) and sequentially tutored group (B) at 55 and 90 dph along with their first tutor (TUT1) and second tutor (TUT2). C) Representative sonograms of novel conspecific birds (NOV) that were used to compare tutee songs with. Syllables copied from TUT1 and TUT2 are indicated by red and blue respectively. Syllables were scored by human observers on a scale from 0 to 3 (0 being the lowest resemblance to a specific tutor syllable and 3 being highest). Syllables that were scored with either 2 or 3 are indicated by red (TUT1) and blue (TUT2) bars. Uppercase roman numerals represent two different sequentially tutored birds and lowercase alphabets represent the different TUT2 used for sequential tutoring. Lower case roman numerals represent different TUT1 used for control and sequentially tutored birds.

Reviewer # 1 suggested to include pictures of the set-up of bird, probe and NMR machine which are now added in D) Picture of an anesthetized bird restrained in a customized bird bed (holder) with the speaker placed into the holder in front of the bird's head. This bird holder is inserted into the probe, and the probe is inserted with the head facing up from the bottom of the NMR machine.

Table 1: The table with supra-threshold clusters was no longer valid with updated thresholds, so we have removed it from the article.

REVIEWERS' COMMENTS:

Reviewer #1 (Remarks to the Author):

The authors have replied satisfactory to all my comments and they re analysed the data more stringently as well as I advised.

I hereby have no further comments except:

- On page 18 reference nr 49 which is 'van der Kant' is written several times as 'Kant' and should be 'van der Kant'

- Legend of Fig 2: Brain regions activated by song in 55-day old birds C-D) and t-values are color coded based on the scale on left.

The scale is on the right? and (A, TUT1 > rest; second tutor song (B, NOV > rest) should be (A, TUT1 > rest) (B, NOV > rest)?

Reviewer #2 (Remarks to the Author):

The authors have addressed all of my questions. I have no further questions. The authors identify the neural correlates of song acquisition from the second tutor in zebra finches. This is an interesting and important study.

Reviewer #3 (Remarks to the Author):

Dear authors, thank you for the thorough response and revision of your manuscript. I have no further comments or questions, and would recommend accepting this manuscript for publication in Communications Biology.

Cover/Rebuttal letter

The changes suggested by reviewer 1 (see below) have all been made. In addition, we changed all figures to boxplots showing all individual data points and clarified biological replicates in the legends in accordance to Comm. Biology's publication policy.

Reviewer #1 (Remarks to the Author):

The authors have replied satisfactory to all my comments and they re analysed the data more stringently as well as I advised.

I hereby have no further comments except:

- On page 18 reference nr 49 which is 'van der Kant' is written several times as 'Kant' and should be 'van der Kant'

- Legend of Fig 2: Brain regions activated by song in 55-day old birds C-D) and t-values are color coded based on the scale on left.

The scale is on the right? and (A, TUT1 > rest; second tutor song (B, NOV > rest) should be (A, TUT1 > rest) (B, NOV > rest)?